# Molecular determinants of nephron vascular specialization in the kidney

David M. Barry[1]*, Elizabeth A. McMillan[1], Balvir Kunar[1], Raphael Lis[1], Tuo Zhang [2], Tyler Lu[1], Edward Daniel [3], Masataka Yokoyama[1], Jesus M. Gomez-Salinero[1], Angara Sureshbabu[4], Ondine Cleaver[3], Annarita Di Lorenzo[5], Mary E. Choi[4], Jenny Xiang[2], David Redmond[1], Sina Y. Rabbany[1,6], Thangamani Muthukumar[4] & Shahin Rafii[1]*

Although kidney parenchymal tissue can be generated in vitro, reconstructing the complex vasculature of the kidney remains a daunting task. The molecular pathways that specify and sustain functional, phenotypic and structural heterogeneity of the kidney vasculature are unknown. Here, we employ high-throughput bulk and single-cell RNA sequencing of the non-lymphatic endothelial cells (ECs) of the kidney to identify the molecular pathways that dictate vascular zonation from embryos to adulthood. We show that the kidney manifests vascular-specific signatures expressing defined transcription factors, ion channels, solute transporters, and angiocrine factors choreographing kidney functions. Notably, the ontology of the glomerulus coincides with induction of unique transcription factors, including *Tbx3, Gata5, Prdm1*, and *Pbx1*. Deletion of *Tbx3* in ECs results in glomerular hypoplasia, microaneurysms and regressed fenestrations leading to fibrosis in subsets of glomeruli. Deciphering the molecular determinants of kidney vascular signatures lays the foundation for rebuilding nephrons and uncovering the pathogenesis of kidney disorders.

[1] Division of Regenerative Medicine, Ansary Stem Cell Institute, Weill Cornell Medicine, New York, NY 10065, USA. [2] Genomics Resources Core Facility, Weill Cornell Medicine, New York, NY 10065, USA. [3] Department of Molecular Biology, University of Texas Southwestern Medical Center, Dallas, TX 75235, USA. [4] Division of Nephrology and Hypertension, Weill Cornell Medicine, New York, NY 10065, USA. [5] Pathology and Laboratory Medicine, Weill Cornell Medicine, New York, NY 10065, USA. [6] Bioengineering Program, DeMatteis School of Engineering and Applied Science, Hofstra University, Hempstead, NY 11549, USA. *email: dbarry553@gmail.com; srafii@med.cornell.edu

Endothelial cell (EC) specialization in each organ is essential for executing tissue-specific functions[1–3]. Kidneys have a unique vasculature to regulate blood pressure (BP), maintain electrolyte homeostasis and pH, and govern red blood cell production[4–7]. Molecular pathways that determine the structural and functional properties of each blood vessel domain in the kidney, as well as the intrinsic and extrinsic cues that enable adaptation of vessels to these tasks are unknown.

Within each human kidney reside approximately one million nephrons, each consisting of a glomerulus and a system of highly distinct tubules. Blood enters the kidney through the renal artery, branches into the interlobular and arcuate arteries that eventually feed into the glomerular capillaries through the afferent arteriole and exit it through the efferent arterioles[8]. The glomerulus is a tuft of fenestrated capillaries, podocytes, and mesangial cells that allow low-molecular weight substances including ions, water, glucose, and nitrogenous waste to pass from the blood into the Bowman's space. The ultrafiltrate of plasma is then transported to multiple segments of the nephron in series including the proximal convoluted tubule, the loop of Henle, the distal convoluted tubule, and the collecting duct. The descending vasa recta (DVR) and ascending vasa recta (AVR) blood vessels run parallel with the loop of Henle. The vasa recta slow the rate of blood flow to maintain an osmotic gradient required for water reabsorption. To execute these complex functions, the kidney vasculature has acquired adaptive structural and functional specialization often referred to as vascular zonation[9]. The intrinsic and extrinsic cues that enable adaptation of the kidney vessels to these tasks are not defined.

The heterogeneity of the kidney vasculature is a major hurdle for the generation of kidney organoids in vitro. Kidney organoids provide avenues for studies of kidney development, disease, and regeneration. Although organoids of the kidney have provided significant insights into kidney physiology and disease, they are mainly limited to the study of developmental processes. Kidney organoids are largely avascular, preventing further maturation and the majority of filtration processes that occur in the kidney. In vivo, the regression of vascular development greatly hinders the organogenesis of the kidney[10]. Similarly, kidney organoid protocols are hampered from building complete, mature, and functional nephrogenic structures in the absence of proper vascularization of the nephron. Current efforts to vascularize organoids have utilized methods such as microfluidic chips[11] or implanting human kidney organoids into the mouse kidney as capsules[12]. Although several studies have recently mapped morphological mechanisms of vascular development in the kidney[13,14], the molecular basis for many of these processes is still largely unknown.

Here, we use bulk and single-cell RNA sequencing (RNA-seq) of the kidney vasculature over a broad range of developmental and adult stages of organogenesis to allow the discovery of distinct transcriptomic signatures associated with different kidney vascular beds. This approach unravels the progression of molecular pathways that specify kidney vascular heterogeneity. We identify several genes—including transcription factors and their putative targets that define the ultrastructural, phenotypic, operational, and functional identity of the glomerular capillaries (GC) and the surrounding vascular network. We additionally perform vascular-specific loss of function experiments on the transcription factor *Tbx3*, a gene that is expressed robustly in glomerular ECs. *Tbx3* regulates a transcriptional network that defines glomerular EC specification and function. In this study, our data uncovers how the transcriptional ontology of the vasculature regulates nutrients and waste in the kidney to sustain chemical and vasomodulatory homeostasis.

## Results

**Molecular profiling of kidney ECs**. To decipher the heterogeneity of the kidney vasculature, we performed comparative transcriptomic analysis of the vasculature of the kidney to that of lungs, liver, and heart at different stages of murine development (Fig. 1a). Each organ was dissected from embryonic stages (E) E13, E14, E15, E16, and E17, postnatal stage (P) P4, and adult mice and dissociated into single cells. We isolated the EC fraction by fluorescence-activated cell sorting (FACS) using fluorescently-conjugated CD31 antibody (Supplementary Fig. 1a, b; Supplementary Data 1).

Affinity propagation clustering (APC)[15,16] of the transcriptomic dataset showed the adult and postnatal vascular expression patterns of each organ to be distinct from those of embryonic stages (Fig. 1b). Given the homogeneity of kidney embryonic vascular gene expression, we considered the kidney early embryonic stages (E13-E16) as a single class and sought to determine how vascular expression patterns change from embryonic to adult stages. In embryonic stages, 657 genes were significantly upregulated and were enriched for growth-related pathways (Supplementary Fig. 1c; Supplementary Data 2). APC analyses showed adult stages to cluster separately and away from each other and from embryonic stages from their respective organs. In the adult kidney, 283 genes were significantly upregulated and were enriched in pathways relating to small molecule, water, and amino acid transport (Supplementary Fig. 1d; Supplementary Data 2).

Thus, gene expression programs distinct to the kidney are turned on early in development to promote growth, morphogenesis, and specification. A wave of genes is differentially induced at various stages of development. At the onset of birth, new sets of genes are induced to promote kidney-specific vascular functions, including upregulation of transporters and metabolism programs, setting the stage for the specialization of kidney vasculature function in postnatal stages. Collectively, we show that kidney vascular heterogeneity diversifies perinatally and throughout adulthood.

**Kidney ECs are heterogeneous and tightly zonated**. To determine how kidney vasculature acquires specialized functions, we dissected kidney EC heterogeneity throughout development. Because kidney-specific vascular genes are induced during late gestation to adult stages (Fig. 1b and Supplementary Fig. 1c, d), we performed single-cell RNA-seq (scRNA-seq) on kidney vascular ECs from fetal E17, perinatal P2, P7 and adult stages (Fig. 1c).

CD31$^+$CD45$^-$Podoplanin$^-$ non-lymphatic ECs were purified by FACS (Supplementary Fig. 1b). Single-cell isolates were then processed for digital droplet scRNA-seq (ddSEQ). We sequenced 5936 cells (Supplementary Data 3), including 922, 1000, 917, and 3097 single ECs for E17, P2, P7, and adult kidneys, respectively. Filtering for contaminating epithelial and perivascular cells reduced the dataset to 4552 cells (Supplementary Data 4). Raw data was normalized and the effects of cell cycle and mitochondrial and ribosomal content were scaled[17]. Dimensionality reduction identified 7 major vascular clusters (Fig. 1d, e), as marked by the expression of pan-vascular EC gene *Cdh5* (Fig. 1f). Clusters were labeled according to known markers and expression validations via protein and RNA staining (Fig. 1g-n, Supplementary Data 5). Cell types identified include the afferent arteriole (AA) and associated large arteries (LA), glomerular capillaries (GC), the efferent arteriole (EA), peritubular capillaries (PTC), the descending vasa recta (DVR), the ascending vasa recta (AVR) and an associated hierarchy of veins and venules leading to the renal vein (V), and embryonic capillaries, which we have designated as vascular progenitors (VP) (Fig. 1o).

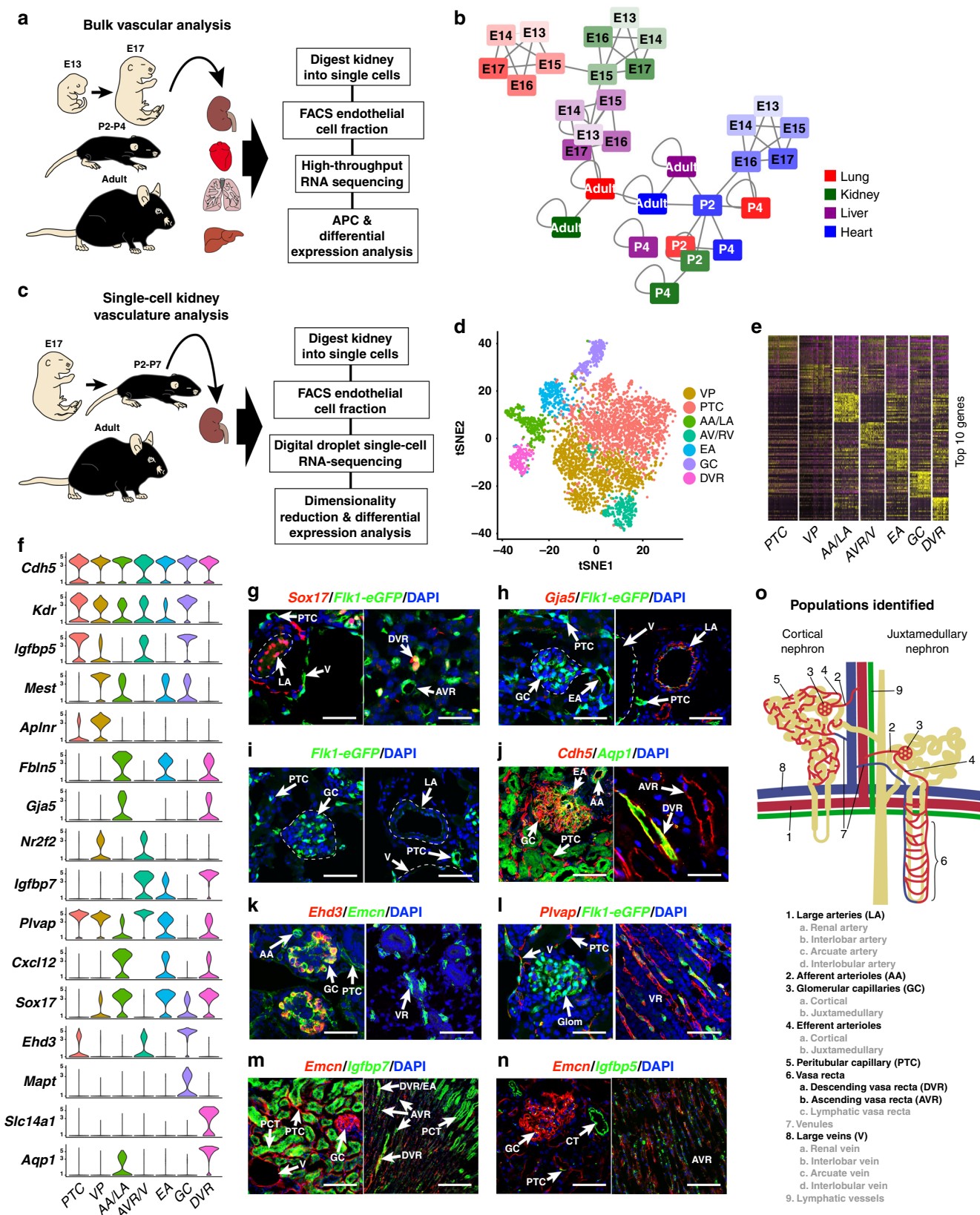

**EC subsets in kidney execute defined vascular functions.** For each population, we selected one or two of the specific genes amongst the top differentially expressed genes and validated the presence of protein expression in various EC populations. Most types of ECs identified did not manifest unique markers. Instead, each vessel displayed markers that were unique to two or more types of vessel. Arterial vessels express markers, including *Sox17* (Fig. 1f, g), *Gja5* (Fig. 1f, h), *Cxcl12*[18], low levels of *VEGFR2* (*Kdr,*

**Fig. 1** RNA sequencing analysis of kidney vascular endothelium throughout development. **a** Diagram denoting the workflow to sequence the bulk transcriptome of the vasculature throughout development. **b** Affinity propagation clustering of each stage. Edge lengths are proportional to Euclidean distances. Stages are color-coded according to the organ. **c** Diagram denoting the workflow to sequence the transcriptome of the vasculature at single-cell resolution. **d** Clustering of single-cell RNA expression according to a reduced dimensionality (t-SNE) for endothelial cells isolated from the kidneys of E17, P2, P7, and adult mice. VP, vascular progenitor; PTC, peritubular capillary; AA/LA, afferent arteriole/large arteries (pre-glomerular); AVR/V, ascending vasa recta/venous blood vessels; EA, efferent arteriole; GC, glomerular capillaries; DVR, descending vasa recta. **e** Heat map denoting genes enriched in each single-cell cluster. **f** Violin plots of normalized scRNA expression profiles of kidney endothelial cells. **g** Staining validation of *Sox17* enriched in arteries in E15 kidneys. Scale bar 50 μm. **h** Staining validation of *Gja5* in arteries except for the efferent arteriole in E15 kidneys. Scale bar 50 μm. **i** *Flk1-eGFP* reporter showing lower expression of *VEGFR2* in arteries. Images were taken at the same exposure. Scale bar 50 μm. **j** *Aqp1* staining in adult human kidney showing enrichment in afferent/pre-glomerular arterioles, and the descending vasa recta. Endothelial cells were marked with VE-cadherin (*Cdh5*) staining. Scale bar 100 μm. **k** FISH staining validation of *Ehd3* showing enrichment in glomerular capillaries in E15 embryos. VR, Vasa recta. Scale bar 50 μm. **l** Staining validation of *Plvap* in peritubular capillaries, veins, and the ascending vasa recta in E15 kidneys. VR, vasa recta. Scale bar 50 μm. **m** Staining validations of *Igfbp7* in the descending and ascending vasa recta in adult kidneys. Endothelial cells were stained with Endomucin (*Emcn*). PCT, proximal convoluted tubule. Scale bar left 100 μm, right 200 μm. **n** Staining validations of *Igfbp5* in glomerular capillaries, peritubular capillaries, and the ascending vasa recta. CT, convoluted tubule. Scale bar 100 μm. **o** Illustration of known vascular subtypes which were identified through ddSEQ. Vascular subtypes not identified are grayed in the text below.

*Flk1*) (Fig. 1f, i), and extracellular matrix proteins, including *Fibulin 5, Elastin, Fibulin 2, Collagen 18a1,* and *Laminin 3* (Fig. 1f, Supplementary Data 5). *Gja5* is expressed in the DVR and all pre-glomerular arteries, including the AA, while absent in the EA[19] (Fig. 1f, h). The DVR traffic urea and water in the medulla of the kidney through the membrane transporters *Slc14a1* and *Aqp1*, respectively (Fig. 1f, j)[20]. The GC can be identified by a variety of specific markers including *Mapt and Ehd3*, a known glomerular EC marker[21] (Fig. 1f, k). Venous ECs were identified through expression of *Nr2f2* (Fig. 1f). The PTCs share several markers with venous EC—including *Plvap* (Fig. 1l) —but were *Nr2f2* negative. They were identified by expression of *Igfbp5*, a marker shared with GC but not with venous ECs (Fig. 1f, n). The AVR was discriminated through presence of *Igfbp7* but lack of *Igfbp5* expression. The PTCs were joined with a seventh population of EC unique to E17-P7 stages. Vessels within this population were *Nr2f2*+ and manifested high expression of *Aplnr* and *Mest* (Fig. 1f). Gene expression signatures were not obtained for lymphatic vessels or larger hierarchies of arteries or veins. Thus, these gene expression signatures enabled us to uncover the differentiation profiles of LA and AA pooled together, GC, EA, PTC, DVR, and a pool of AVR and V (Illustrated in Fig. 1).

**Vascular heterogeneity arises from vascular progenitor cells**. To discern how vascular heterogeneity arises in the kidney, ECs were ordered according to pseudotime[22] (Fig. 2a–c, Supplementary Fig. 2a). The earliest capillary cells correspond to VPs and were labeled by the apelin-receptor gene, *Aplnr* (Fig. 2d). Quantifications of *Aplnr* fluorescent staining validated its restriction to capillaries and veins[10,23] (Fig. 2h, i). Pseudotime trajectory predicts two branch-points from early VPs. At the first branchpoint, pre-glomerular large arteries (AA and LA) branch from the VPs (early generic capillaries) (Fig. 2a, Supplementary Fig. 2a). This transition is marked by downregulation of vein and VP signatures, such as *Plvap, Aplnr, Nr2f2*, (Fig. 2b, d, e, Supplementary Fig. 2b) and upregulation of gene signatures found in (but not specific to) pre-glomerular arteries (AA/LA), such as *Gja5, Fbln5, Jag1, Sox17*, and *Cxcl12* (Fig. 2b, f, Supplementary Fig. 2c, d). Accordingly, arteries and veins are the first vascular subtypes, which build up from capillaries at E13 before other vascular structures[13]. At the second branchpoint, VPs transition into GCs (*Lpl*+, *Ehd3*+, *Sema5a*+, etc.) (Fig. 2c, g) in association with EAs or the DVR (*Slc14a1*+, *Aqp1*+) (Fig. 2c, Supplementary Fig. 2f, g), which is successive to the juxtaglomerular efferent arteriole. Remaining VP capillaries then mature into PTCs in the kidney cortex, downregulating VP markers including *Aplnr* and *Nr2f2*, while maintaining partially unique markers *Igfpb5* and

*Plvap* (Fig. 2c-e, Supplementary Fig. 2b, e). Large veins may emerge from VP capillaries and maintain expression of venous markers, including *Nr2f2*+ and *Cryab*+ ECs (Fig. 2b, Supplementary Fig. 2b).

To unravel the plasticity of kidney EC subtypes, we pursued lineage-tracing clones of VPs. We utilized R26R-Confetti mice expressing four fluorescent reporter genes and crossed them to the vascular-specific and inducible Cdh5(PAC)-CreERT2 strain[24]. Pregnant mice were pulsed with tamoxifen at E11, when ureteric buds emerge. ECs began to express single clones of green (GFP), cyan (CFP), yellow (YFP), and red fluorescent proteins (RFP) at E13 (Supplementary Fig. 2h). By E18, vasa recta, veins, arteries, and glomerular vascular subtypes were labeled with only one fluorescent protein (Fig. 2j, k). Pseudotime analysis predicts the EAs branch off progenitors in tandem with GCs while pre-glomerular arteries and arterioles develop independently. Notably, a single arteriole was found to protrude from developing GCs marked with the same fluorescent protein while pre-glomerular arteries were labeled separately (Fig. 2j). Progenitors of cortical PTC displayed heterogeneous mixture of differentially labeled and diverse ECs. Because progenitor capillaries were the first to be labeled, VP ECs may have migratory potential and do not interact during development, leading to intermingling of confetti colored cells. Collectively, specified ECs appear to remain within their structures after they have developed and do not draw a large degree of plasticity to migrate and become other vascular subtypes.

Vasa recta and glomeruli are reported to grow from PTCs in the cortex between E14-E15 stages[13,25]. Bulk RNA-seq (Fig. 3a) and staining validations of the vasculature at E15 (Fig. 1g–o) show genes specific to vascular subtypes become expressed at E14–E15 stages. Peritubular capillaries (the vascular progenitors) were found to constitute the primary vascular plexus in the cortex of the kidney at E13 stages (Fig. 2l). Following vascular specification, transcripts defining vascular subtypes gradually become upregulated over the course of development, peaking at adult stages. Therefore, vascular heterogeneity in the kidney is specified at E14–E15 stages where they branch off VP capillaries in the cortex. As ECs branch off, they lose and gain several key markers and mostly sustain the phenotype of their designated vascular subtypes.

**Changes in kidney expression dynamics during development**. The vasculature in adult kidneys differ from other organs by modulating trafficking of amino acids, water, and ion transport proteins, while embryonic stages are defined by specific induction of cell growth and morphogenic pathways (Supplementary Fig. 1c, d). Hence, we sought to determine how gene expression

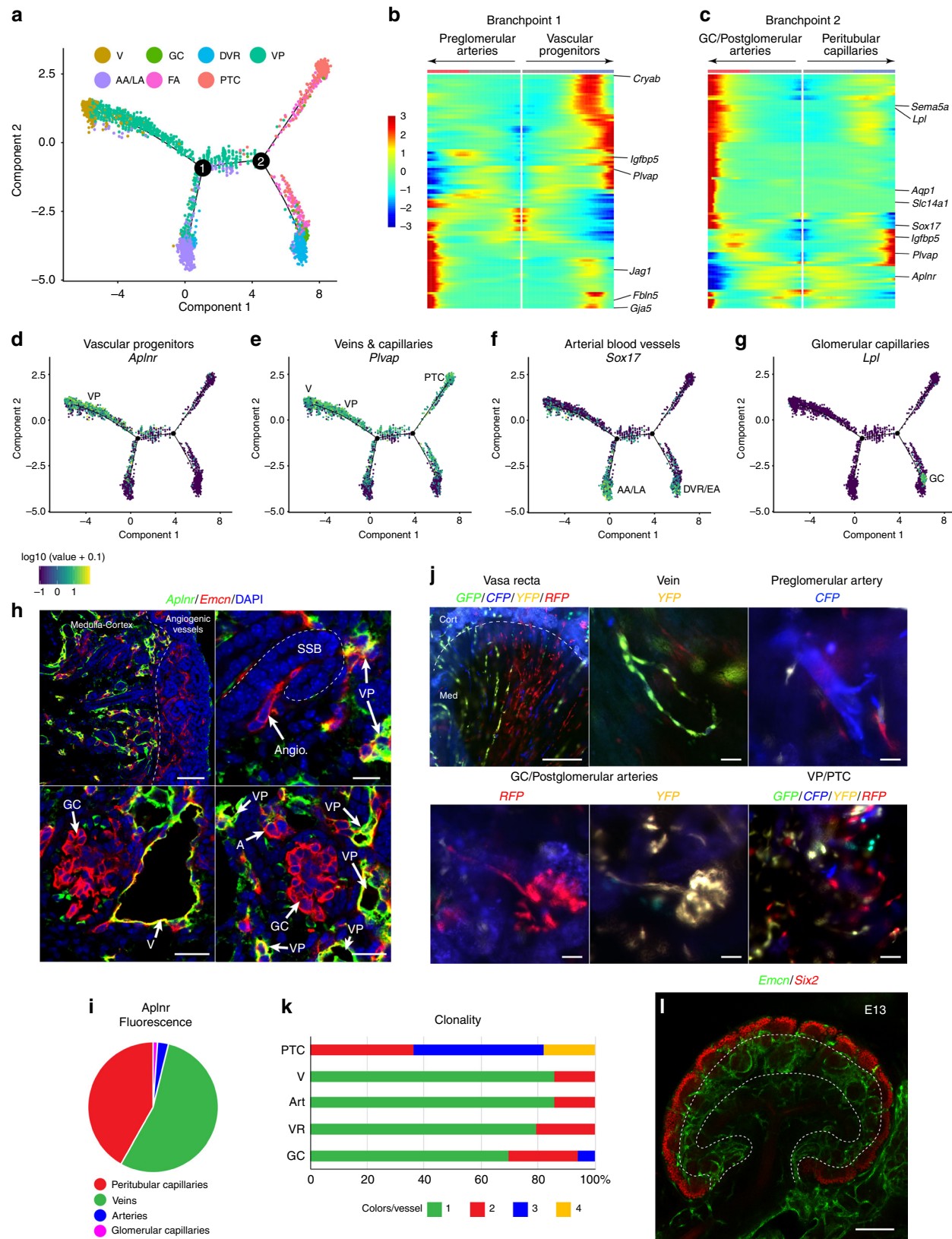

associated with transcription factors, solute transporters, and growth factors change within each vascular subpopulation over the course of development. From bulk RNA-seq of the kidney vasculature, we first annotated differentially expressed genes in the kidney vessels at each stage of development.

We defined 617 transporter proteins, 1314 transcription factors, and 2032 secreted proteins, from the transporter classification database (http://www.tcdb.org/), the DBD transcription factor prediction database[26], and the human protein atlas (https://www.proteinatlas.org/). We find sets of genes expressed

**Fig. 2** Analysis of vascular heterogeneity development. **a** Pseudotime trajectory of vascular differentiation in the kidney. VP, vascular progenitor; PTC, peritubular capillary; AA/LA, afferent arteriole/large arteries (pre-glomerular); AVR/V, ascending vasa recta/venous blood vessels; EA, efferent arteriole; GC, glomerular capillaries; DVR, descending vasa recta. **b** Heat maps denoting genes that become differentially expressed as pre-glomerular arteries branch from vascular progenitor cells. Notable genes of venous peritubular progenitor capillaries (*Cryab, Igfbp5, Plvap*) and arteries (*Jag1, Fbln5, Gja5*) are shown on the right. **c** Heat maps denoting genes that become differentially expressed as glomerular capillaries (GC) and postglomerular arteries branch from embryonic progenitor capillaries which mature into peritubular capillaries. Notable genes of glomerular capillaries (*Sema5a, Lpl*), postglomerular arteries (*Aqp1, Slc14a1*), and peritubular capillaries (*Igfbp5, Plvap*) are on the right. **d–g** Pseudo-time trajectory plots denoting the expression of genes enriched in particular vascular clusters. Plots include *Aplnr* in vascular progenitor cells (VP) (**d**), *Plvap* in veins (V), VPs, and peritubular capillaries (PTC) (**e**), *Sox17* in afferent arterioles/large pre-glomerular arteries (AA/LA) and descending vasa recta/efferent arterioles (DVR/EA)(**f**), and *Lpl* in glomerular capillaries (GC)(**g**). **h** Immunofluorescent staining of the apelin receptor (*Aplnr*) in E17 kidney. Endothelial cells were stained with endomucin (*Emcn*). SSB, s-shaped body; VP, vascular progenitor; Angio., angiogenic vessel; GC, glomerular capillary; V, vein; A, arteriole. Scale bars: first panel 40 μm, second panel 5 μm, third and fourth panel 10 μm. **i** Pie chart denoting mean fluorescent intensity of *Aplnr* antibody staining in peritubular capillaries, veins, arteries, and glomerular capillaries. n = 3, average of 5 frames of view. **j** R26R-Confetti E18 mouse kidneys cut in half sagittally after tamoxifen induction at E11. GC, glomerular capillary; PTC, peritubular capillary. Scale bars: first panel 100 μm, second to sixth panel 5 μm. **k** Bar graph denoting the number of fluorescent reporters found in identified vascular structures. n > 10 for each structure. Art, Arteries; VR, Vasa Recta. **l** E13 mouse kidney showing the primary vascular plexus exists as generic capillaries (the vascular progenitor cells) before subvascular specification at E14-E15 stages. Dotted lines outline the cortex and medulla. Endothelial cells are stained with endomucin (*Emcn*) and the outer cortex of the kidney is denoted by *Six2* staining of nephron progenitors. Scale bar 100 μm.

in the early embryo (E13–E16) are highly enriched for transcription factors (hypergeometric test p-values range from 1.1E-4 to 3.1E-5), but not for secreted proteins and transporters. Transcription factors are downregulated at E17 (Fig. 3b), with stabilization of their expression (hypergeometric test $p = 0.89$) thereafter. Notably, the bulk of gene expression of vascular genes in postnatal and adult stages, are dominated by transporters (adult hypergeometric test $p = 5.1.E-18$) and secreted proteins (adult hypergeometric test $p = 0.027$) (Fig. 3b). From the P4 to the adult stage, transporters specific to a wide variety of substrates that are not expressed in other organs are enriched in the kidney vasculature (Supplementary Data 5). We also detected an increase in total numbers of transcripts in these stages compared to embryonic stages (Supplementary Fig. 3a). Notably, membrane transport proteins were transcribed only after birth and their expression are augmented steadily to adulthood. Therefore, transport proteins may be induced in ECs in response to the environmental stimulation and dietary changes.

**Specialized capillaries regulate filtrate reabsorption.** As the glomerular filtrate passes through different segments of the kidney tubule, selective reabsorption of the filtrate is returned to the systemic circulation through the PTC. We investigated whether vasculature zonation accommodates selective filtrate reabsorption and secretion. We compared tubular epithelium and the surrounding stroma through sequencing the mRNA of the non-endothelial fraction ($CD31^-VEcadherin^-CD45^-Podoplanin^-$) and non-lymphatic endothelial fraction ($CD31^+VEcadherin^+CD45^-Podoplanin^-$). In kidney ECs, 34 transporters were uniquely expressed, including a variety of calcium and potassium channels as well as transporters for phospholipids, glucose, and amino acids (Supplementary Fig. 3b).

We integrated scRNA-seq data to investigate whether the expression of transporter proteins in kidney vessels is also zonated to guide selective filtrate reabsorption. We find membrane transporters to exhibit both punctate and ubiquitous expression patterns. For instance, *Slc25a3* and *Slc25a5* that transport mitochondrial phosphate and adenine, respectively, are ubiquitously expressed across the kidney vasculature (Supplementary Fig. 3c). Select transporters were also found to be specific to endothelium, such as *Slc9a3r2*, a protein involved in sodium absorption (Fig. 3e–g). However, many membrane transport proteins identified were expressed within distinct populations of the kidney vasculature (Fig. 3e–m, Supplementary Data 5) supporting the notion that transporters tailor the function of surrounding nephrogenic zones.

Temporal analysis of transporters specific to vessels reveals that most membrane transport proteins are induced at E14–E15 stages, while being upregulated from E17 to adult stages (Fig. 3d). The GCs specifically express two of the voltage gated channels, *Kcnj5* and *Scn7a*, which are both linked to hypertension[27,28] (Fig. 3h, i). Pre-glomerular arterial ECs (AA/LA) express genes, such as *Slc8a1* and *Kcnn4* that pump calcium (Fig. 3e). This is consistent with the role calcium plays in regulating the contraction and relaxation of smooth muscles surrounding the arterial endothelium[29]. Efferent arterioles present high levels of *Slc6a6*, which mediates membrane transport of taurine, modulating osmoregulation, membrane stabilization, antioxidation, and the conjugation of bile acids[30,31] (Fig. 3j, k). *Slc6a6* gene ablation predisposes mice to streptozotocin-induced diabetic nephropathy[32]. The DVR cluster selectively express urea transporter *Slc14a1* and *Aqp1*, which plays a central role in urea and water transport, respectively (Fig. 3e, l, m)[20]. The organic anion transporter *Slco4a1* mediates $Na^+$-independent uptake of thyroid hormones and bile acids in the apical membrane of the proximal convoluted tubule[33,34]. We find this transporter is expressed in venous blood vessels (AVR/V) (Fig. 3e). Thus, organized distribution of chemical transporters to various kidney ECs may regulate electrolyte balance and homeostasis of various molecules in the blood.

**Kidney EC-epithelial crosstalk evolves over development.** Kidney epithelial cells and the surrounding stroma crosstalk to mediate development and homeostasis of the kidney[10]. The vasculature establishes a niche that through the secretion of specific angiocrine signals regulate specific developmental and homeostatic functions. Complete functional vascularization is required for progressing the development of the kidney and kidney organoids[12]. However, the mechanism by which zone-specific vascular niche angiocrine signals sustain nephron homeostasis is unknown.

To address this, we show that multitudes of factors are secreted by different vascular subtypes in different ECs from E17 to adult stages, (Fig. 3c). Notably, GCs supply *Fgf1*, *Vegfa*, Notch-ligand *Dll4*, neuronal guidance cue *Sema5a*, and Wnt antagonist *Dkk2* (Supplementary Fig. 3d). Contaminating podocyte or mesangial doublets were not detectable in the glomerular EC cluster (Supplementary Fig. 3d). During developmental progression, arterial ECs produce *Tgfβ2*, *Ltbp4*, and *Pdgfβ*, while VPs elaborate *Igf1* and *Igf2* with their expression shifting to veins and arteries, respectively (Supplementary Fig. 3e). Regulatory molecules of IGF signaling zonate throughout the vasculature of the adult nephron

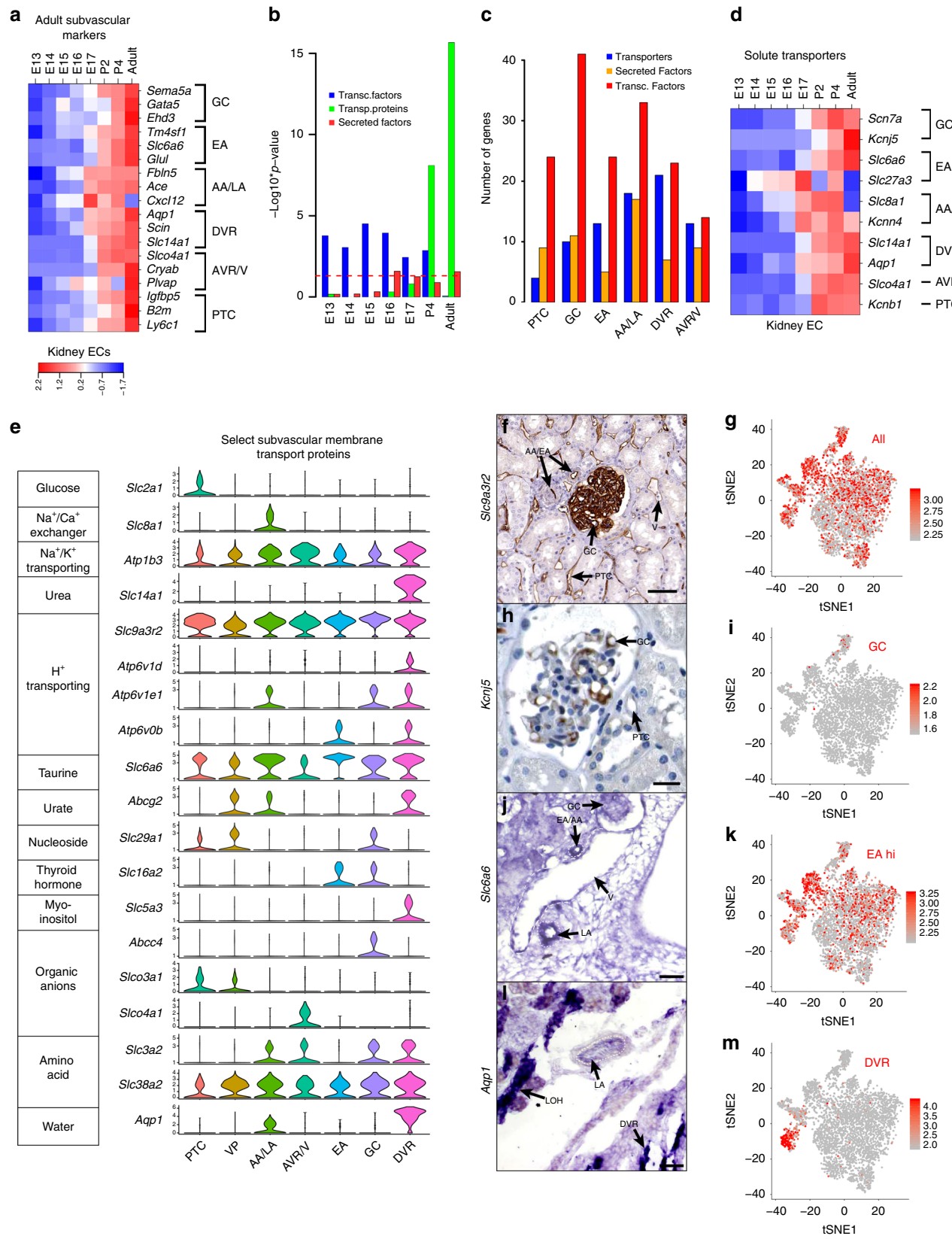

(Supplementary Fig. 3e). Thus, a network of angiocrine and autocrine factors manifest restricted transcription among defined kidney vascular zones. In addition, we have developed a database to find what types of vessels in the kidney express specific secreted angiocrine factors that contribute to morphogenesis and patterning of nephrons.

**Fig. 3** Transport and transcriptional ontology is zonated in vascular subtypes in the kidney. **a** Heatmap relating expression profiles (z-scores) of 3 top highly enriched genes to each vascular cluster across 7 developmental stages. VP, vascular progenitor; PTC, peritubular capillary; AA/LA, afferent arteriole/large arteries (pre-glomerular); AVR/V, ascending vasa recta/venous blood vessels; EA, efferent arteriole; GC, glomerular capillaries; DVR, descending vasa recta. **b** Hypergeometric p-values indicating enrichment of transcription factors, transporters, and secreted proteins in the lists of genes differentially expressed between the kidney and the heart, liver, and lungs at each stage of development. Red dotted line denotes *p*-values >0.05. **c** Numbers of transporters, growth factors, and transcription factors found to be differentially expressed in each vascular cluster. **d** Heatmap relating expression profiles (*z*-scores) of the top 1 or 2 highly enriched transporters to each vascular cluster across 7 developmental stages. **e** Violin plots showing the normalized expression level of representative solute transporter genes across the 7 vascular zones of the kidney. Genes shown here were chosen arbitrarily. *Y*-axis is log scale-normalized read count. **f, g** Human protein atlas image and tSNE showing an example of a pan-endothelial transporter, *Slc9a3r2*, in the kidney vasculature. Scale bar 40 μm. AA, afferent arteriole; EA, efferent arteriole; GC, glomerular capillary; V, vein; PTC, peritubular capillary. Patient 2184, image available from v18.1.proteinatlas.org. https://www.proteinatlas.org/ENSG00000065054-SLC9A3R2/tissue/kidney#img. **h, i** Human protein atlas image and tSNE showing an example of a glomerular capillary transporter, *Kcnj5*, in the kidney vasculature. Scale bar 10 μm. Patient 1767, image available from v18.1.proteinatlas.org. https://www.proteinatlas.org/ENSG00000120457-KCNJ5/tissue/kidney#img. **j, k** In situ hybridization and tSNE showing an example of a transporter, *Slc6a6*, that is pan-endothelial and differentially highly expressed in arterioles. Scale bar 20 μm. **l, m** In situ hybridization and tSNE showing an example of a transporter, *Aqp1*, that differentially expressed in the DVR and large arteries (LA); loop of Henle (LOH). Scale bar 20 μm.

**Transcription factors delineate the kidney vasculature.** Specific transcription factors are enriched in various stages of developing kidney vasculature (Fig. 3b). The bulk of these genes are Hox family of transcriptional regulators (Supplementary Data 2). However, in postnatal stages, many of these genes are down-regulated and other specific sets of transcription factors are induced (Supplementary Data 2). Notably, many different transcription factors are associated with distinct vascular subtypes (Supplementary Fig. 3f, Supplementary Data 6), the majority of which are upregulated during late embryogenesis (E17). Hence, specific sets of transcription factors are induced during later stages of development to promote additional vascular specialization or function.

To identify transcription factors that may be important for the development of the kidney vasculature, we utilized the single-cell regulatory network inference and clustering (SCENIC) method[35]. We identified 216 transcription factor regulons enriched in our dataset, which we then binarized and clustered with a supervised analysis to obtain lists of transcription factors with differential activity amongst each vascular subtype (Fig. 4a, Supplementary Fig. 3f). As expected, arteries and veins manifested an enrichment of *Sox17* and *Nr2f2* regulons, respectively. The GCs produced the largest and most unique combination of transcription factors. Many components of the AP-1 heterodimer such as *Fos*, *Jun*, and *Atf* were enriched (Fig. 4a). The AP-1 pathway is generally enriched during cellular stress and infection[36], suggesting that glomerular ECs may be sensitive to cellular stress during processing and tissue digestion. The glomerulus expressed several transcription factors known to be involved in stem cell differentiation and tissue morphogenesis, including *Tbx3*, *Gata5*, *Prdm1*, *Irf8*, *Zbtb7a*, *Klf4*, *Maff*, and *Klf13*. These gene transcriptional regulatory networks that can be investigated to further our understanding of kidney physiology and manipulated to enhance the construction of kidney tissues in vitro.

**Tbx3 contributes to glomerulus morphogenesis and function.** To elucidate mechanisms underlying glomerular development and function, we focused on transcription factors with restricted and abundant expression and regulon activity (Fig. 4a), such as *Tbx3* in GCs and EAs (Fig. 4b). Immunofluorescent staining in human kidney tissue confirmed that *Tbx3* protein is restricted to ECs in GCs and EAs (Fig. 4c).

To uncover the role of Tbx3 in glomerular specification and function, we conditionally ablated *Tbx3* expression in ECs by employing the Cre/LoxP system with *Cdh5-Cre* (*VE-cadherin-Cre*) to generate *Tbx3^ΔEC^* mice. The *Tbx3^flox^* allele loses 4.6 kb of genomic DNA encompassing the 5′UTR promoter, the

transcriptional start site and first exon of *Tbx3* in the presence of *Cre recombinase*[37]. Targeted homozygous deletion of the *Tbx3* allele in ECs using the vascular-specific *VE-cadherin-Cre*[38] manifested clearly noticeable morphogenic defects in subsets of the glomeruli in *Tbx3^ΔEC^* mice, but not in the control *Tbx3^flox/flox^* or *Tbx3^flox/+^;Cdh5^Cre^* mice (Fig. 4d, Supplementary Fig. 4a). In 4-month-old mice, most glomeruli appeared normal although 18% developed microaneurysms, 5% of glomeruli exhibited hypoplasia, and 3% of glomeruli became fibrotic (Fig. 4d, e). Affected glomeruli manifested significantly dilated capillaries and arterioles (Fig. 4d–f).

Urinalysis revealed *Tbx3^ΔEC^* mice to have higher levels of urea, protein, and salt, suggesting aberrant nephron filtration or reabsorption (Fig. 4g–l). Consistent with most glomeruli appearing normal, serum panel showed few signs of kidney failure (Supplementary Fig. 4b–g). Salt homeostasis, blood urea nitrogen, creatinine, and albumin levels remained normal in *Tbx3^ΔEC^* serum, with normal glomerular filtration rate (Supplementary Fig. 4k). Four-month-old *Tbx3^ΔEC^* kidneys did not have significantly fewer ECs and did not show signs of kidney inflammation or apoptosis as marked by cleaved *Caspase-3* and *CD45* staining, respectively (Supplementary Fig. 4h, i). No significant defects were found in major blood vessels in other organs (Supplementary Fig. 4j). Transmission electron microscopy (TEM) revealed a fraction of GCs with microaneurysms, significantly fewer fenestrations, and surrounded by deformed podocyte foot processes (Fig. 4m, Supplementary Fig. 4l). Thus, *Tbx3* maintains the structural organization of glomerular capillaries.

To determine whether *Tbx3* plays a physiological role in BP homeostasis in adult mice, we measured systolic BP in control and *Tbx3^ΔEC^* mice. Systolic BP in mice lacking endothelial *Tbx3* was lower compared to that of control mice (Fig. 4n). To assess whether this was due to defects in the renin–angiotensin system, levels of renin, angiotensinogen (*Agt*), and angiotensin-converting enzyme (*Ace*) were measured in the kidney, lung, and liver respectively by qPCR. Renin levels were found to be higher in the absence of *Tbx3* in the vasculature of the kidney, while *Agt* and *Ace1* transcript levels remained the same (Fig. 4o–q). Therefore, *Tbx3* regulates a putative transcriptional program that suppresses or balances blood pressure via regulation of renin in the kidney.

To uncover the mechanism by which *Tbx3* mediates glomerular vascular development and function, ECs were isolated from control and *Tbx3^ΔEC^* kidneys and processed for RNA sequencing (Supplementary Fig. 4m; Supplementary Data 7). Differential expression analysis of glomerular-specific genes between control and *Tbx3^ΔEC^* kidney vasculature revealed an

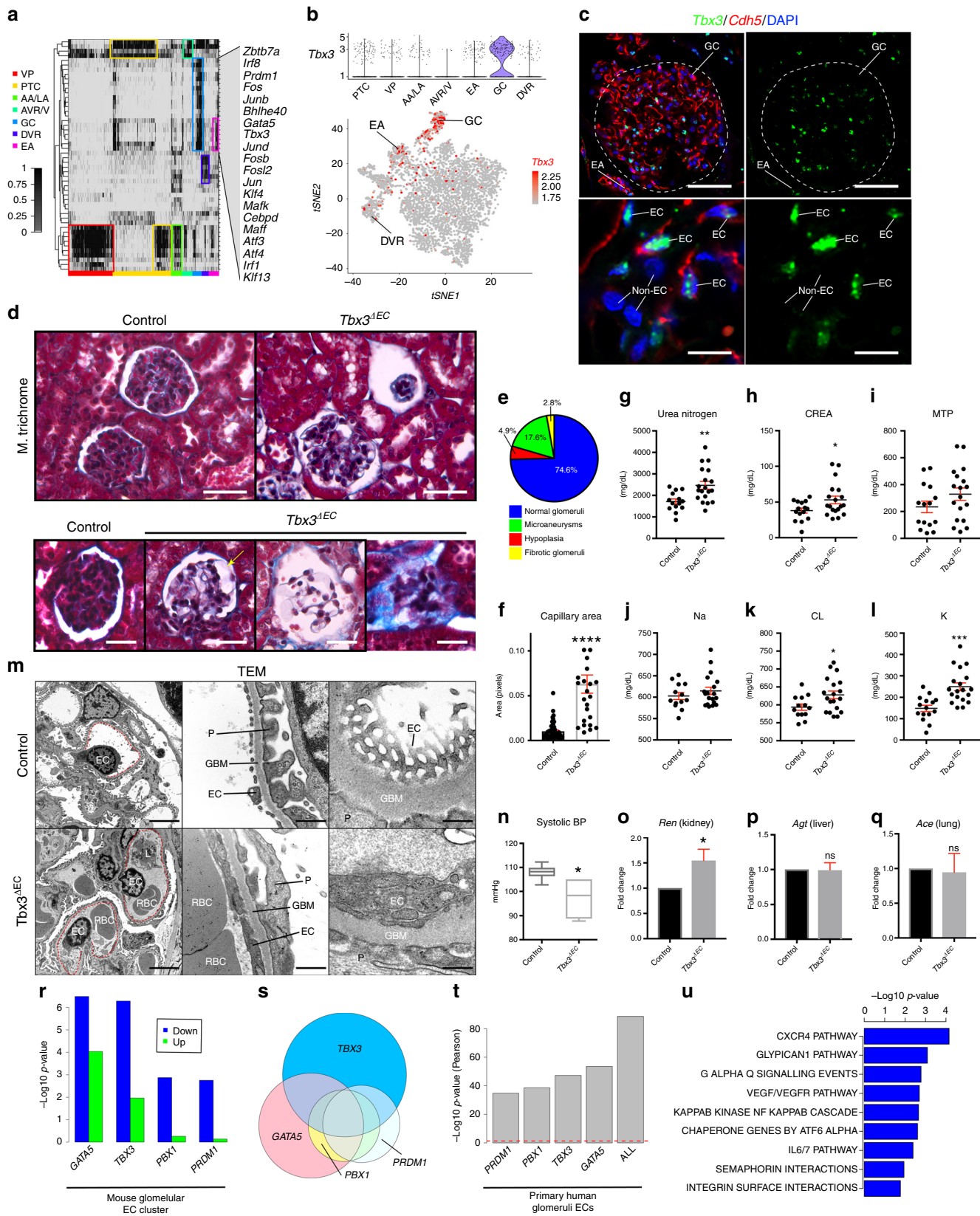

array of transcripts that were differentially expressed after *Tbx3* loss in mice (Supplementary Fig. 4n). Downregulated genes include the solute transporter *Slc44a2*, transcription factors *Gata5*, *Klf4*, and *Smad6*, and proteins that mediate adhesion or the cytoskeleton, including *Rhob* and *Itga3*. The expression of *Ehd3*, the most abundant and specific gene in GCs (Fig. 1k, 3a), was also downregulated in the absence of *Tbx3*. Genes that increased in the absence of *Tbx3* include those necessary for

**Fig. 4** *Tbx3* is necessary for glomerular morphogenesis and specification. **a** Heat map denoting SCENIC results. VP, vascular progenitor; PTC, peritubular capillary; AA/LA, afferent arteriole/large arteries (pre-glomerular); AVR/V, ascending vasa recta/venous blood vessels; EA, efferent arteriole; GC, glomerular capillaries; DVR, descending vasa recta. **b** Violin plot and tSNE showing normalized *Tbx3* expression. **c** Immunofluorescent staining of *Tbx3* and VE-cadherin (*Cdh5*) in adult human glomerular endothelial cells. EC, endothelial cell; Non-EC, non-endothelial cell. Scale bar 50 μm, inset scale bar 10 μm. **d** Masson's trichrome staining in control and *Tbx3^{ΔEC}*. Yellow arrow show microaneurysms. Scale bar 50 μm. **e** Pie chart for percent of glomeruli possessing microaneurysms, hypoplasia, or fibrosis. **f** Quantification glomerular capillary area. $n = 3$ mice, 5 frames/kidney. ***$p < 0.0001$ unpaired Student $t$-test. Error bars, standard error of the mean. **g–l** Urine analysis (control $n = 15$, *Tbx3^{ΔEC}* $n = 19$). **g** Urea nitrogen, **h** creatinine (CREA), **i** micro total protein (MTP), **j** sodium (Na), **k** chloride (CL), **l** potassium (K). Normalized to volume. * $p < 0.05$, **$p < 0.01$, ***$p < 0.001$ unpaired Student $t$-test. Error bars, standard error of the mean. **m** Transmission electron microscopy (TEM) of control and *Tbx3^{ΔEC}*. Red dotted line outlines glomerular capillary lumens. P, podocyte; GBM, glomerular basement membrane; EC, endothelial cell; RBC, red blood cell; L, leukocyte. Scale bar 2 μm, inset scale bar 500 nm. **n** Systolic blood pressure ($n = 6$ mice/group). *$p < 0.05$ unpaired Student $t$-test. Center line = median. Bounds of boxes: the first to third quartile. Whiskers highlight quartile from minimum or maximum. Error bars, standard error of the mean. **o–q** qPCR of Renin (*Ren*), Angiotensinogen (*Agt*), and Angiotensin-converting enzyme (*Ace*) transcripts (kidneys, liver, and lung, respectively). $n = 4$ mice each group. *$p < 0.05$ unpaired Student $t$-test. Error bars, standard error of the mean. **r** Hypergeometric test $p$-values ($-1*\log_{10}$) for overlap between mouse glomerular genes (scRNA sequencing) and genes down (blue) or up (green) regulated upon overexpression of the indicated transcription factors. EC, endothelial cell. **s** Euler plot illustrating overlap between glomerulus genes downregulated with overexpression with transcription factor. **t** Hypergeometric test $p$-values ($-1*\log_{10}$) for overlap between human glomerular-specific genes (human glomerular endothelial cells) and genes downregulated upon overexpression of the indicated transcription factors (ALL = all 4 TF's). EC, endothelial cell. Red dotted line denotes $p$-value >0.05. **u** Pathway enrichment analysis after over expression of *Tbx3, Gata5, Prdm1*, and *Pbx1*.

morphogenesis, (*Spry4*, *Rap1b*, and *Pcdh17*), transcription factors (*Irf1*, *Prdm1*, *Elf1*, and *Stat3*), and the growth factor *Fgf1*. While glomeruli represent a small fraction of the kidney vasculature, there was a significant overlap between genes that are differentially expressed in the bulk endothelial population of *Tbx3^{ΔEC}* and genes specifically expressed in the glomerular cluster (hypergeometric test $p = 6.7E-9$; Supplementary Fig. 4o).

To identify potential targets of *Tbx3*, either *Tbx3* or a control vector was overexpressed in human umbilical vein endothelial cells (HUVECs), representing a generic-like vascular bed, and mRNA was sequenced (Supplementary Data 8). There was a significant overlap between *Tbx3*-regulated genes in HUVECs and glomerular-specific genes (hypergeometric test $p = 7.1E-10$, Supplementary Fig. 4o). When we compared glomerular genes that were upregulated in *Tbx3^{ΔEC}* mouse kidney ECs and Tbx3-regulated genes in HUVECs, we find a high concordance with glomerular genes identified through scRNA-seq (hypergeometric test $p = 4.8E-5$; Supplementary Fig. 4o). Hence, *Tbx3* is a very potent and specific mediator of glomerular gene transcription and identity.

**Tbx3, Gata5, Prdm1, and Pbx1 repress and specify GC genes.** Other transcription factors were found to have enriched expression or activity in GCs in addition to *Tbx3* (Fig. 4a), prompting the hypothesis that additional factors might cooperate with *Tbx3* to establish glomerular function. To determine whether transcription factors cooperate in GCs, we focused on additional factors with differential expression, including *Tbx3*, *Gata5*, *Prdm1*, and *Pbx1*. Each gene was stably over-expressed in HUVECs and mRNA was sequenced for each condition (Supplementary Data 8). Log$_2$ fold-changes were calculated relative to an empty-vector control (EV) to find genes regulated by each transcription factor. We compared the lists of genes that change in response to stable overexpression of each transcription factor and found that a significant proportion of genes that were downregulated correspond to genes found within the GC cluster (Fig. 4r). The overlap, however, was not nearly as prominent for the sets of genes that are upregulated. Thus, the identified zone-specific transcription factors may be performing as repressors and possibly collaborate to regulate a common set of glomerular-specific targets.

We next sought to determine if transcription factor over-expression, either alone or in combination, is enough to approximate glomerular fate in HUVEC cells. As a positive

reference control, we used glomeruli isolated from a deceased-donor human kidney that was not transplanted[39] (Supplementary Fig. 4p, q). $CD31^+VE$-cadherin$^+CD45^-$ ECs were FACS-sorted from glomeruli and processed for RNA sequencing. To enable direct comparison between human glomeruli and transcription factor overexpression datasets, all datasets were normalized as a fold-change value relative to the HUVECs expressing an empty lentivirus vector. We selected the top 10 significantly up or downregulated genes in response to each overexpression condition and compared expression patterns to human glomeruli. The top differentially regulated genes in response to each transcription factor overexpression exhibited similar expression patterns and phenocopied expression patterns in human glomeruli (Supplementary Fig. 4r). Expression patterns in response to *PRDM1* and *PBX1* very highly overlapped with that of *TBX3* and *GATA5*, despite *PRDM1* and *PBX1* being more ubiquitously expressed, though significantly higher in GCs (Fig. 4s). We directly compared overlap among the expression of glomerular-specific genes in the human glomerulus and in response to each transcription factor. We find that genes modified in response to overexpression of each transcription factor very highly overlap with genes specifically expressed in the human glomerulus (Fig. 4t). We next overexpressed all 4 transcription factors simultaneously in HUVECs and found the highest overlap with the human glomerular expression patterns, indicating these factors function additively in dictating glomerular specificity. Hence, overexpression of each transcription factor promotes a signature that renders HUVECs more like human glomeruli and opens avenues towards designing systems to conditionally reprogram cultured cells into human glomeruli.

To identify pathways downstream of these transcription factors, we did pathway enrichment analysis on GC-specific genes regulated by these factors (Fig. 4u). Numerous vascular maintenance pathways, including those found to be enriched in the glomerulus were downregulated. These include the *CXCR4* and *VEGFR2* signaling pathways, semaphorin interactions, and integrin signaling. This data prompted the hypothesis that each of these four proteins, controling a transcriptional hierarchy through which glomerular-specific targets guide vascular functions, are selectively downregulated to execute various glomerular functions. Accordingly, in *Tbx3^{ΔEC}* mice, we observe subsets of glomeruli manifesting hypoplasia, microaneurysms, and fenestration loss, potentially leading to aberrant BP homeostasis. Notably, BP was decreased and there was no indication of vasculopathy in other organs that could induce secondary hypertensive changes in

the kidney vasculature (Supplementary Fig. 4j). We conclude that *Tbx3*, along with other factors turns on the expression of pathways that confer the unique vascular functions of GCs. Dysregulation of these genes could lead to defects associated with developmental patterning, EC rarefaction, EC adhesion and contraction, and solute transport.

## Discussion

To decipher the molecular determinants regulating intra-kidney vascular network diversity, we employed ddSEQ single-cell RNA sequencing[40–42]. We identified 6 discrete non-lymphatic vessels in the kidney vasculature. Although several important blood vessels are missing, including lymphatic vessels, we focused primarily on unraveling the signature of glomerular ECs, and broadly the arteries, veins and capillaries. Each vascular domain displays unique membrane transport proteins, regulators of transcription, growth factors, and endocrine hormone binding proteins that serve known and unknown functions in nephron development, filtrate reabsorption, and blood pressure homeostasis.

Kidney vascular specification begins at E14 while waves of organotypic kidney vascular genes are induced at the onset of birth, in perinatal stages, and during maturation into adult vessels. Genes that are unique to the kidney vasculature relative to other organs are robustly expressed after birth. Compared to the early developmental stages, several gene signatures in the adult kidney vasculature are unique. We propose that this unexpected transition of vascular specialization could be due to the physiological stress on the kidney after birth. Notably, membrane transport proteins that are important for the kidney's vasculature function are not upregulated until after birth, indicating that expression of these transporters may be dependent on stimuli not present during gestation. Alternatively, the transcriptome of the postnatal and adult stages may dramatically differ from the embryonic stages of the same organ through the loss of developmental and mitotic transcripts and the transition to transcripts the kidney needs to function.

How do kidney blood vessels, specifically the capillaries, acquire this complex heterogeneity? One hypothesis suggests extrinsic cues turn on transcription factors driving vascular specialization. To test this, we parsed out genes specific to the kidney and specific to zones of the vasculature. We find that GCs primarily express factors that are suppressors of transcription. Pathways repressed by each of these transcription factors correspond to the same pathways that are enriched in the GCs found using scRNA-seq. This creates a conundrum as to how genes become specifically expressed in the glomerulus, but also downregulated where *Tbx3/Gata5/Prdm1/Pbx1* is expressed. It is conceivable that prior to glomerular specification, the ECs activate generic vascular specification programs with broad vascular gene profiles. *Tbx3, Prdm1, Gata5*, and *Pbx1* may then be recruited to prune gene expression and fine-tune the specialized functions of the glomeruli. Each factor may also modulate other suppressors thereby activating GC genes. Alternatively, each gene may solely function to modulate or balance perinatal and adult kidney-specific processes, including BP homeostasis or glomerular filtration dynamics.

As an example of intrinsic transcriptional suppression regulating GC functions, we focused on *Tbx3*, as it is robustly represented in the GCs and represses transcription of particular genes in a variety of diverse, yet specific, tissues[43]. $Tbx3^{\Delta EC}$ mice developed phenotypes similar to capillary endotheliosis—glomerular swelling and loss of fenestrations—in subsets of the glomeruli. Differential expression analysis among pooled *Tbx3*-deleted mouse kidney endothelial cells, *TBX3*-over-expressing

HUVECs, and genes enriched in the murine and human glomeruli revealed that a network of genes that regulate vasodilation/constriction, adhesion, and solute transport that may contribute to hypoplasia, microaneurysms, and loss of fenestrations in $Tbx3^{\Delta EC}$ mice.

A network of developmental genes is also altered in Tbx3-deleted ECs, including *Gata5* and *Ehd3*. Both genes have been shown to be necessary for the integrity of glomerular endothelium. Mice lacking *Gata5* and *Ehd3* develop glomerular lesions and kidney failure[21,44]. $Tbx3^{\Delta EC}$ mice also have inversely correlated phenotypes to mice that lack endothelial *Gata5*, including aberrant BP and renin signaling, suggesting that both transcription factors may balance BP in the kidney and possibly arterial vessels in other organs. *Tbx3* in arterioles, and glomerular endothelium, may suppress genes that allow crosstalk to juxtaglomerular cells, therefore regulating renin secretion, glomerular filtration rate, and BP homeostasis. *Tbx3* may also regulate glomerular filtration dynamics by stimulating crosstalk to podocytes and mesangial cells under certain conditions. *Tbx3* was not found to be expressed in blood vessels in other tissues except for the lungs, which may explain, why phenotypes were centralized primarily in the kidneys. *Tbx3* may also function redundantly with its paralogue *Tbx2*, possibly masking additional phenotypes. Additional studies are necessary to prove these hypotheses and to gain a better understanding of kidney vascular zonation, development, and function.

Our composite of endothelial-specific signatures bring us closer to engineer zone-specific kidney endothelium. Notably, each vascular bed expresses a combination of secreted factors and transcriptional regulators that define zonated vascular fate. By supplying angiocrine factors, ECs instructively orchestrate tissue healing and regeneration during lung, liver, and bone marrow injury[45–47]. Generating kidney-specific endothelium may facilitate approaches to rebuild kidneys. Our dataset compiles an array of genes that could be used to engineer kidney-specific vascular endothelium. Furthermore, this knowledge will lay the foundation for identifying extrinsic environmental cues that confer kidney vessels with their unique region-specific structural and functional activities.

## Methods

**Cell digestion and FACS sorting**. To isolate adult endothelial cells from mouse kidney, liver, heart, and lung, mice were injected intravitally with 25 µg of anti-VE-cadherin-AF647 antibody (clone BV13, Biolegend) retro-orbitally in 6–8-week-old male C57BL/6J mice under anesthesia 10 min before they were killed and the organs harvested. For cell sorting, organs were minced and incubated with Collagenase A (25 mg ml$^{-1}$) and Dispase II (25 mg ml$^{-1}$) at 37 °C for 20–30 min to create a single-cell suspension. Cells were filtered through a 40-µm filter immediately before counter staining. The single-cell suspension was first blocked with an Fc-quenching antibody before antibody staining with anti-mouse CD31-Alexa Fluor® 488 (102414, Biolegend), anti-mouse CD45-Pacific Blue™ (103126, Biolegend), and anti-mouse Podoplanin-PE/Cy7(127412, Biolegend). Embryonic tissues were dissected and processed through the same antibodies. Following staining, cells were processed for FACs sorting.

**Immunofluorescent staining**. Tissues were fixed overnight in 4% paraformaldehyde at 4 °C. The following day, organs were washed in PBS and cryoprotected in 30% sucrose overnight. Tissues were then embedded in Tissue-Tek O.C.T. Compound and sectioned at 10 µm on a cryostat. Frozen sections were washed with PBS to remove O.C.T. Antigen retrieval was performed on select slides using a pressure cooker. During retrieval, slides were soaked in citrate buffer (EMS, buffer B cat. # 62706–11 for cytoplasmic stains and buffer A catalog number 62706–10 for nuclear stains). The tissues were permeabilized in PBS with 0.1% Triton X-100 and blocked for 1 h at room temperature in 5% normal donkey serum. Primary antibody incubations were done at 4 °C. Slides were then washed in PBS, incubated in secondary antibody for 1 h at room temperature. Slides were then washed in PBS and mounted using Prolong Gold Mounting Medium containing DAPI. Images were obtained using an A1R Nikon confocal microscope and a Zeiss LSM 710 Confocal Microscope. The following antibodies were used: chicken anti-GFP (for Flk-GFP; Aves, GFP-1020, 1:500), goat anti-Cx40/Gja5 (Santa Cruz, sc-20466, 1:100), rabbit anti-Collagen IV (Millipore, AB756P, 1:400), goat anti-Nrp1 (R&D

Systems, AF566, 1:100), rat anti-Plvap (BD Pharmingen, 550563, 1:100), goat anti-Sox17 (R&D Systems, AF1924, 1:100), rat anti-Endomucin (Santa Cruz, sc-65495, 1:100), rabbit anti-Aquaporin1 (Biorad, MCA2100, 1:100), rabbit anti-Tbx3 (Abcam, ab99302, 1:100), rabbit anti-Aplnr (Protein Tech, 20341–1-AP, 1:100), goat anti-Igfbp5 (R&D Systems, AF578, 1:100), goat anti-Igfbp7 (Abcam, ab129302, 1:100), rabbit anti-Six2 (Protein Tech, 11562–1-AP, 1:100).

**In situ hybridization**. Fixed E15.5, E18.5, or P5 kidneys were dehydrated to 100% ethanol and embedded in paraffin before sectioning using a microtome. Sections were de-paraffinized in xylene, then rehydrated to PBS before being treated with 15 μg/mL proteinase K for 15 min and fixed in 4% PFA/PBS. Slides were then washed and incubated with a pre-hybridization buffer for 1 h at room temperature before being hybridized with the specific probes at 1 μg/mL overnight at 65 °C. The following day, slides were washed in 0.2x SSC then transferred to MBST, and blocked with 2% blocking solution (Roche) for at least 1 h at room temperature. Slides were then incubated with anti-Dig alkaline phosphatase-conjugated antibody (Roche, 1:4000) overnight at 4 °C. Next day, slides were washed 3x in MBST and NTMT before incubating with BM purple (Roche) for a color reaction. After the color reaction, slides were fixed with 4% PFA and mounted using Permount mounting solution. Images were taken using a Zeiss Axiovert 200M scope and a DP-70 camera from Olympus.

Fluorescent in situ hybridizations were performed following the same procedure up to SSC washes. Following the washes, slides were transferred to TNT and treated with 0.3% $H_2O_2$ for 30 min. Slides were then washed again in TNT and blocked in 1% blocking buffer (Perkin Elmer) for 1 h at room temperature. Slides were then incubated with anti-Dig peroxidase (Roche, 1:500), rat anti-PECAM, and rat anti-Endomucin antibodies overnight at 4 °C. Next day, slides were washed in TNT 3x before incubating with TSA Fluorescein Amplification Reagent (1:50 in Amplification Diluent, Perkin Elmer) for 15 min. Slides were washed in TNT following TSA incubation, incubated with goat anti-rat Alexa Fluor 555 for 2 h at room temperature, and then incubated in DAPI. Slides were washed in TNT and mounted using Prolong Gold Mounting Medium. The slides were imaged using an A1R Nikon confocal microscope

**Masson's trichrome staining**. Masson's trichrome staining was performed by Histoserv, Inc. To fix the tissues, following euthanasia, mice were perfused with 25 mls PBS then 10 mls of 4% PFA/PBS through the left ventricle of the heart. The right atrium of the heart was severed to accommodate bleeding. Kidneys of the mice were then additionally fixed overnight in 4%PFA/PBS overnight at 4 °C. The following day, the kidneys were washed with PBS then stored in 40% ethanol before being shipped to Histoserv, Inc.

**Transmission electron microscopy**. TEM was carried out by the Weill Cornell Medicine Electron Microscopy Core Facility per their standard protocols.

**Lentivirus protocol**. The lentiviral vectors used to overexpress TBX3 (pLV[Exp]-Hygro-hPGK > hTBX3[NM_016569.3]), GATA5(pLV[Exp]-Hygro-hPGK > hGATA5[ORF024149]), PRDM1(pLV[Exp]-Hygro-hPGK > hPRDM1 [NM_001198.3]), and PBX1(pLV[Exp]-HygromPGK > mPbx1[ORF039780]) in our study was constructed and packaged by VectorBuilder (Cyagen Biosciences).

**Human umbilical vein endothelial cell culture**. HUVECs were isolated from umbilical cords at the New York Presbyterian Hospital. The permission and approval for obtaining discarded or left over umbilical cords were obtained from institutional review board (IRB) at Weill Cornell Medicine. The IRB deemed the studies on HUVECs exempt from the requirement of informed consent. The primary HUVECs cultured on plates coated with gelatin in media consisting of M199 (Sigma, M4530), 10% FBS (Omega Scientific, FB07), 50 μg ml$^{-1}$ endothelial mitogen (Alfa Aesar J65416), and 100 μg ml$^{-1}$ heparin (Sigma, H3393).

**Clinical pathology analysis**. Urine analysis and serum analysis panels were carried out by the Memorial Sloan Kettering Cancer Center laboratory of comparative pathology per their standard protocols. Urine was collected using metabolic chambers. Analysis of urine was normalized to the volume that was produced. Serum was collected retroorbitally via a heparin-coated capillary after mice were anesthetized with isoflurane.

**Glomerular filtration rate**. In all, 100 mg of Inulin-FITC was boiled into 5 ml 0.85% NaCl then filtered into a Bio-Spin gel column. Mice were anesthetized with isofluorane, then retroorbitally injected with the Inulin-FITC solution (2 μl/g bw). Mouse tails were clipped and blood was collected at 0, 3, 5, 7, 10, 15, 35, 56, and 75 min post injection via a capillary coated with heparin. The blood was spun down then plasma was diluted 1:10 in 0.5 M HEPES. The concentration of FITC was then measured using a Spectra Max photometer (485 excitation and 538 emission). GFR was calculated in GraphPad Prism using a two-phase exponential decay function.

**Blood pressure measurement**. Between the hours of 8–10 a.m., systolic blood pressure was measured using an IITC Life Science tail cuff plethysmography blood pressure system.

**Animal husbandry**. All animal experiments were performed under the approval of Weill Cornell Medicine Institutional Animal Care and Use Committee (IACUC), New York, NY. The breeding and maintenance of animal colonies abided by the guidelines of the IACUC of Weill Cornell Medical College, New York, New York, USA. All experimental procedures followed the IACUC guidelines. Genotyping was carried out in the laboratory or the tails were sent to Transnetyx (transnetyx.com). To compare the phenotypes among different mouse genotypes, sex- and weight-matched littermates were used. The study used 4-month-old male mice. Tbx3 mice (Tbx3$^{tm3.1Moon}$)[37], provided by Anne Moon (Weis Center for Research), were crossed with Cdh5-Cre (Tg(Cdh5-cre)7Mlia)[38], from Luisa Arispe (UCLA) to produce Tbx3$^{flox/flox}$; Cdh5$^{Cre}$ mice (referred to as Tbx3$^{ΔEC}$), and maintained as homozygous mice. The Cre allele was maintained in a heterozygous stage after it was bred in (Cre/+). Male littermates were used for all assays comparing control and Tbx3$^{ΔEC}$ mice. Flk1-eGFP mice (Kdrtm2.1Jrt)[48] were kindly provided from Ondine Cleaver at UT Southwestern Medical center. R26R-Confetti mice (Gt (ROSA)26Sortm1(CAG-Brainbow2.1)Cle) purchased from Jax and crossed to Cdh5 (PAC)-CreERT2 (Tg(Cdh5-cre/ERT2)1Rha) mice from Ralph Adams were kindly donated from Jason Butler at Weill Cornell Medicine.

**Human tissue data**. Human kidneys used for glomerular endothelial cell isolation or histology were obtained as medical waste from a deceased-donor human kidneys that were not transplanted. The deidentified, discarded human kidneys used for research are not considered as "human subject research" as per the standard NIH definition. Hence no IRB approval of the protocol was required.

**Single-cell RNA-seq analysis**. Cells were harvested from E17, P2, P7, Adult murine kidneys and were digested into single cells. A single-cell suspension was loaded into the Bio-Rad ddSEQ Single-Cell Isolator (BioRad, Hercules, CA) on which cells were isolated, lysed and barcoded in droplets. Droplets were disrupted, and cDNA was pooled for second strand synthesis. Libraries were generated with direct tagmentation followed by 3′ enrichment and sample indexing using Illumina BioRad SureCell WTA 3′ Library Prep Kit (Illumina, San Diego, CA). Pooled libraries were sequenced on the Illumina NextSeq500 sequencer at pair-end read (R1: 68 cycles, sample index: 8 cycles and R2: 75cycles). Sequencing data were primarily analyzed using the SureCell RNA Single-Cell App in Illumina BaseSpace Sequence Hub. In particular, sequencing reads were aligned to the mouse mm10 reference genome using STAR aligner;[49] cell barcodes were used to separate reads from different cells, and unique molecular identifiers (UMI) were used to remove duplicate reads that were actually derived from the same mRNA molecule. A knee plot was generated based on the number of UMI counts per cell barcode in order to identify quality cells separating from empty beads or noise, and a raw UMI counts table for each gene in each cell was then prepared. The raw counts table was fed into Seurat version 2.0.1 for clustering analysis[17]. Cells that have between 200 and 2500 detected genes were kept for downstream analyses. This filtering step was used to filter for high-quality single cells. Epithelial cells and perivascular cells were filtered based on the expression of Cdh1/Epcam/Cdh16 and Pdgfrb, respectively. The resulting data was log-normalized in Seurat. To mitigate the effects of cell cycle heterogeneity in data, we followed a previously published approach by assigning each cell a score based on its expression of canonical cell phase markers and then regressing these out using Seurat. We also regressed out effects associated with the number of UMIs, mitochondrial content and ribosomal gene content. Principal component analysis (PCA) was performed on the top variable genes determined in Seurat, where the top 13 principal components were selected by choosing the inflection point in the Scree plot and were used for cell clustering and t-SNE projection. t-SNE plots were generated using R ggplot2 package.

**Pseudo-time analysis**. Data was normalized and cells were filtered using tools available in the Seurat package, as described above. Normalized data was converted to an object useable by Monocle V2 in R. Low quality reads were detected as those with a minimum normalized expression of <0.1 and which were expressed in at least 10 cells. The dimensionality of the dataset was reduced with a DDR tree with the number of dimensions set at 13, defined according to the procedure listed above. The number of dimensions were selected by choosing the inflection point in the Scree plot. Cells were then ordered in pseudo-time and trajectories were plotted using Monocle 2.

**SCENIC**. SCENIC[35] was used according to the protocol previously described using the protocol in the SCENIC package in R. Briefly, the transcription factor network was defined based on co-expression and filtered using GENIE3 in R using the GENIE3 and GRNboost packages. Cells were first filtered to be those expressed in at least 1% of cells with a count value of at least 3. Targets for transcription factor regulons were then scored with RcisTarget. Cells were scored based on the activity of the gene regulatory network with AUCell and cells were clustered according to GRN activity with t-stochastic neighbor embedding using only high confidence regulons. Regulon activity was binned according to the activity above the AUC

threshold. A one-way hierarchical cluster was drawn based on binned regulon activity using the stats package in R. Cells were ordered according to clusters defined in the Seurat package described above.

**Isolation of glomeruli from human kidney**. Human kidney was decapsulated uniformly and minced into small fragments with scalpels. These small fragments were then digested with 1 mg/ml collagenase III in RPMI medium at 37 °C for 40 min with mild rotation. The specimens were then gently pressed with a flattened pestle and passed through a 100-μm cell strainer. After washing with complete medium (RPMI + 10% FBS) for three times, glomeruli were collected and centrifuged at $55 \times g$ for 5 min. The emerging suspension contained uniformly decapsulated human glomeruli with minimal disintegration.

**Affinity propagation clustering**. APC was performed according to pipelines published by McMillan et al[15]. Replicate values were collapsed by the median expression value. As input to the algorithm, cells were clustered by the top 20% of the most highly variant transcripts, corresponding to 3331 genes.

**RNAseq normalization pipeline**. As a part of our standard RNA-seq pipeline, Fastq files were quality checked with FastQC (https://www.bioinformatics.babraham.ac.uk/projects/fastqc/) and reads were processed to remove adapter sequences with BBtools (https://jgi.doe.gov/data-and-tools/bbtools/). Reads were aligned to the mm10 mouse genome or the hg38 human genome with STAR v2.5.3a[49]. Aligned files were sorted and indexed with samtools v1.5[50], and count files were generated with HTseq v0.9.1[51]. Counts were imported to R v3.4.0, batch corrected and normalized with the EdgeR package[52]. The limma[53] package was used to calculate differential expression and assign $p$ and $q$-values.

**Pathway analysis**. Genes were associated as being differentially expressed between conditions if the limma derived $p$-value was <0.001 and the $\log_2$ fold change was >1. Gene sets were curated from Broad MSigDB V3 (Kegg, Reactome, GO)[54] or CORUM[55] databases and filtered for gene sets containing between 5 and 200 members. The sets of transcription factors were defined from the transcription factor classification database (http://www.tcdb.org/) and the sets of membrane proteins were defined from the DBD database[26]. Growth factors were curated from the lists of secreted proteins in the human protein atlas. A hypergeometric test was used to calculate enrichment of gene sets in lists of genes and $p$-values were adjusted for false discovery rate.

**Other statistical analyses**. Hierarchical clustering was performed using the stats package in R.

**Reporting summary**. Further information on research design is available in the Nature Research Reporting Summary linked to this article.

## Data availability
All bulk and single-cell expression matrices have been deposited under Gene Expression Omnibus with the accession numbers GSE129005 and GSE137786. Other associated raw data are available in Supplementary Data 1–8. All data are available from the corresponding authors upon reasonable request.

## Code availability
Code for processing single-cell RNA-seq is available in the Supplementary Software.

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

## Acknowledgements
We are grateful to Dr. Anne M Moon for supplying the Tbx3^tm3.1Moon mouse strain. T.M. is supported by the R03 National Institute of Digestive and Kidney Diseases (NIDDK) R03 DK105270 grant. M.E.C. is supported by National Institute of Health (NIH) R01 HL133801, R01 HL060234, R01 HL055330, and R01 HL132198. A.D.L. is supported by National Institute of Heart, Lung and Blood (NHLBI) R01 126913. S.R. is supported by Ansary Stem Cell Institute, Division of Regenerative Medicine at Weill Cornell Medicine (WCM), and National Institute of Health (NIH) Director's Transformative award R01 HL128158 and National Heart, Lung, and Blood Institute (NHLBI) HL139056, National Institute of Allergy and Immunology (NIAID) U01 AI17001 and Daedalus fund for innovation Weill Cornell Medicine and New York State Stem Cell Science (NYSTEM) Consortium Award and Avalon GloboCare Corp training grant and Lisa Dean Moseley Foundation. This research was supported in part by National Cancer Institute (NCI) Grant NIH T32 CA203702.

## Author contributions
Conceptualization by D.M.B., Sina R., T.M. and Shahin R.; Methodology by D.M.B., E.A.M., J.X., A.D.L. and R.L.; Software by E.A.M. and T.Z.; Formal analysis by D.M.B., E.A.M., T.Z., B.K. and D.R.; Investigation by D.M.B., B.K., R.L., T.L., E.D., M.Y., J.M.G. and A.S.; Writing by D.M.B and Shahin R. reviewed by all other authors; funding by Shahin R., M.E.C. and O.C.

## Competing interests
S.R. is a co-founder and non-paid consultant to Angiocrine BioScience, San Diego CA. The spouse of M.E.C. is a cofounder and shareholder, and serves on the Scientific Advisory Board of Proterris, Inc. All other authors have no conflicts of interest.

## Additional information

**Peer Review Information** *Nature Communications* thanks Virginia Papaioannou and other anonymous reviewer(s) for their contribution to the peer review of this work. Peer reviewer reports are available.

