## [Peer Review File · Nature Communications]

Reviewers' comments:

Reviewer #1 (Remarks to the Author):

In the manuscript entitled: Molecular determinants of glomeruli vascular specialization in the kidney, Rafii and colleagues provide comprehensive analysis of developing glomerular microvasculature through analysis of single cell RNAseq from isolated blood (non-lymphatic) endothelial cells from the kidney. In addition, the authors provide evidence for a role of *tbx3* in normal glomerular capillary formation and maturation. The paper is well-presented and provides important information and datasets to advance studies and development of therapeutics for kidney diseases – many of which affect glomeruli and their vasculature.

The following comments are for consideration by the authors:

1. Title – should be glomerular instead of glomeruli. However, the paper provides more data than just glomerular microvasculature – so would consider changing title to reflect more comprehensive data provided (which is valuable).
2. Perinatal heterogeneity – perhaps not published but not surprising given all the post-natal differentiation that occurs in adjacent structures (tubular transporters etc. aren't expressed or functional at birth but develop post-natally) – so tubular filtrate and what is reabsorbed by endothelium is very different. Could soften this statement in the introduction.
3. *Tbx3* endothelial-specific knockout mice exhibit an interesting and somewhat unexpected phenotype. Typically, lower *gfrs* are associated with higher blood pressures – as are higher renin levels (if angiotensin 1, 2 levels are higher as a result of higher renin – one would expect a higher blood pressure). Do the authors suspect a sodium wasting phenotype in the mice to explain the *bp* findings (that are counterintuitive)? This would not be glomerular but downstream - typically quite distal in the nephron. A concentrating defect in the urine might also suggest this – are the mice polyuric? Is *Tbx3* also expressed in *avr*? Given the *cre*-driver is expressed in > glomerular endothelial cells, this might help explain the phenotype.
4. Would remove the comment about hyperemic kidneys suggesting perfusion abnormal – soft phenotype and not diagnostic of perfusion issue – although they do look different – suggesting fixation and flushing of vasculature was not equivalent between both.
5. It is likely there are more than 6 subpopulations of endothelial cells in the kidney – could soften this conclusion in the discussion. Findings might be limited by number of cells, stages, and might have missed some subpopulations.
6. Any data to compare glomerular endos to liver sinusoidal endothelium or choriocapillaris (comparison to other published, available datasets is fine)? Given similarities between glomerular endos and some of these other fenestrated vascular beds.

7. Some additional details about blood pressure measurements and histologic comparisons would be helpful – age of mice, littermates, gender specified. How were the fenestrations quantified? Etc.

Reviewer #2 (Remarks to the Author):

1. Use of the term “evolution” as in the “evolution of pathways” in the introduction and several other places in the text is a questionable word choice. This is not evolution in the darwinian sense of the word. The authors also use the term “ontology”, which may be a better choice.

2. The sentence at the bottom of page 4 is problematic for several reasons. First, inclusion of the word “kidney” in the penultimate line is probably a typo. Secondly, use of the term “Unexpectedly” begs the question of what assumptions led the authors to find this result unexpected? It is not immediately obvious.

3. The sentence at the top of page 10 states that two transitory (fleeting) vascular networks are enriched in the developing cortex and a venous-like vascular bed. This implies knowledge of the location of the populations and sounds like they are distinct. What is the evidence supporting this statement?

4. The sections dealing with the conditional mutation of Tbx3 are lacking in essential details. First, the description of the mutant mice is too sketchy. If the mice from Anne Moon, Luisa Arispe, and Ondine Cleaver have been published, references should be provided and MGI proper allele nomenclature used. A quick check of MGI indicates that there are at least 5 different Cdh5-Cre transgenes and a number of Tbx3 floxed alleles. Without detailed information on the specific alleles used, it is impossible to evaluate the quality of these mutations or to ascertain from this study that they are performing as designed. Pertinent details such as how faithfully the Cre transgene recapitulates endogenous Cdh5 expression and how efficient it is at excision of floxed alleles is essential information. Also, it is stated that the Tbx3 mice are maintained as homozygous (sic) but it is not clear if the Cre transgene is homozygous or segregating. Do the mice examined have one or two copies of the Cre transgene? Did controls include mice carrying the Cre transgene? This is an important control as Cre is known to have effects on its own in some cases.

5. There is no discussion of potential effects of knocking out Tbx3 in EC of other tissues, which presumably would occur. There is mention on page 21 that there is no indication of vasculopathy in other organs but no data is presented. Are the mice generally sick? How thoroughly were the mice examined for additional extra-kidney abnormalities?

6. With respect to the glomeruli in Tbx3 Δ EC mice, the proportion of abnormal glomeruli is fairly low. How does this proportion compare with normal mice and with Cre-only controls? In the measurements of kidney function (Fig 4G-L), do the controls include mice with the Cre transgene? In

fact for all the comparisons using Cre transgenes, Cre-only controls are essential unless this transgene has been previously shown not to affect these measurements.

7. How representative are the TEM pictures and are they taken at random? Do these represent the 75% apparently normal glomeruli or only the abnormal glomeruli? It is important to relate these TEM to the histology to understand how widespread this phenotype is and whether all the glomeruli or only the histologically abnormal ones are affected at the TEM level.

8. At the bottom of page 19, what is the basis for the conclusion that the transcription factors “collaborate” to regulate a common set of targets?

Minor points for correction:

1. There are numerous grammatical errors that need editing.

2. Fig. 1C the E16.5 kidney symbol must be hidden behind the blue square. Both could be visible if it were in front of the square.

3. Fig. 2 What age tissue is the expression validation done on? In general, the age of tissue shown in figures is not stated, e.g. what is the age/developmental stage of the human tissue in Fig. 4C (text page 16) or the mouse tissue in Fig. 4C. It should be clearly stated in the text and figure legends.

4. Reference to Fig. 3D in the middle of page 15 seems to be incorrect as this figure is on solute transporters, not transcription factors.

5. It should be made clear on the bottom of page 16 that Cdh5-Cre and VE-cadherin-Cre are the same thing.

6. Middle of page 18, Figure 4O should be 4Q.

Reviewer #3 (Remarks to the Author):

In this work Barry et al. characterize the heterogeneity of endothelial cell populations in the kidneys. They focus on describing the vascular zonation molecularly and in characterizing the molecular determinants of the endothelial cells heterogeneity. The study, also, elucidates how these molecular differences unfold developmentally.

The main results of the paper are: (1) the definition of a specific molecular profile for 2 early and 6 mature endothelial cell populations in the kidney; (2) the description of when these differences arise

developmentally; (3) the observation that gene regulation mechanisms leading to the glomerular fate is mostly repressive and (4) the characterization of the knock-out phenotype of Tbx3 a transcription factor specific to glomerular endothelial cell.

The first part of the paper has more the characteristics of a resource (Figure 1,2,3), and it is completed by a second part (Figure 4) that aims at a more in-depth experimental characterization starting from few specific findings.

I believe the paper represents an important characterization of the kidney heterogeneity to a level of detail that had escaped previous, less focused, single-cell surveys of the organ (i.e., Quake et al. 2018 "Tabula Muris"). The developmental aspect, in conjunction with the gene regulation one, is particularly interesting as it describes the timing of kidney endothelial specification to unprecedented detail. We would like to recommend this work for publication in Nature Communications as long as the authors make substantial changes to the manuscript, significantly extend and improve the RNA-seq analysis and its statistical rigor, placing more care in the conclusions reached from the data. We also suggest only a few experimental extensions to complete and clarify the data the authors present.

The part of the paper that I am more concerned about is the bulk transcriptomics analysis of the endothelium of different organs as compared with the kidney endothelium (Paragraph "Molecular profiling of kidney endothelium" and Figure 1A-B, S1A-D). There are several problems with this part of the paper:

(1) First and foremost, it seems (the author do not state otherwise) that all the data has no biological or technical replicates, in other words, $N=1$!!! This fact alone takes a heavy toll on the statistical validity of the results, and if really N is equal 1, then no statistical inference can be made from this part of the data.

(2) The comparison between a general "kidney endothelial cell" and the ones from other organs is conceptually flawed, especially if considering the data that the authors have at hand. The authors show, later in the paper, that there are important differences between endothelial cell types in the kidney. So that comparing a pool of different endothelial cells RNA mixed together results in an artefactual cell expression pattern that does not exist in any real cell but is the average of them all. The entire analysis is influenced by this averaging, so statements like "kidney endothelial cells possess distinct expression pattern" are imprecise, the inferences here is on a heterogeneous pool difficult to analyze and interpret and therefore at the mercy of artefactual results (e.g. a case of Simpson's paradox). We also note that in this analytic scenario even a slight change in the composition of the subtypes might influence the results of the analysis.

(3) There is a significant gap between the data interpretation provided by the authors and what the data presented is suggesting. The PCA analysis shown in figure 1 is overinterpreted and the authors stretch too many conclusions out of it. Despite the fact that the authors try to lead the eye of the

reader by drawing circles around points the differences that they comment on are not so clear from the data. Furthermore, we noticed that the sample coordinates on PC1 are not zero centered and have an unusual range, are the authors using some other sample in the analysis and not showing it on the plot? Also being the percentage of variance explained by PC1 very high (small in PC2) and the values of PC1 all positive lead me to conclude that the difference between the data point is just linear scaling of the expression profile. This would be in sharp contrast to what the authors claim and suggest a mere difference in detection levels or cell size.

(4) The authors ignore the availability of at least a dozens of single-cell datasets in the literature that incidentally includes endothelial cells of one or more organs, and that could be used to make a better analysis than the one allowed from this bulk experiments.

In summary, I believe the evidence in this part of the paper is confusing, and the analysis does not add any relevant information to the main story. I would therefore strongly recommend the authors to completely remove this analysis from the paper and start directly from the single cell RNA-seq data. The alternative would be the substitution of their analysis with an improved or different dataset that can provide more compelling evidence of their claims.

The single-cell data collected is adequate to most of the claims made by the authors. Maybe it could have even be used to make further or more precise claims.

Notably, the authors do not propose a model for the change in the heterogeneity of the endothelial pool and specification observed during development. This cell state transition is an important finding in the economy of the paper and the authors should try to complete its analysis and description. Can the data be used to build a model of the lineage determination starting from the 2 early types and ending in 6 mature types? For example, the data should allow determining the structure of the lineage either by branching pseudotime analysis (monocle 3), optimal transport analysis (Waddington OT) or RNA velocity analysis (velocityto) to try to come up with a model of this developmental progression.

Coming up with a putative lineage branching structure would also allow a more detailed analysis of the specification of endothelial cells. We strongly encourage the authors to at least propose a putative qualitative model of this developmental process based on the evidence available. What is the most likely progeny of each adult clusters? Is it possible to determine which transitions are allowed? Which cells are proliferating and which one postmitotic? If the authors did not develop these analyses because they are convinced that the data is insufficiently detailed or high quality to fit this model, they should consider sampling more cells.

Finally, the two early subtypes (cluster 1 and 2) define transcriptionally should be anchored to cell populations in the tissue by means of RNA-scope or smFISH (better than IH because some of the

genes may not be translated at a higher level yet). In this way, the gene expression patterns can be connected to the histology and cell morphology offering a more precise picture of what these populations are.

Furthermore, concerning the finding that a perinatal specification and postnatal maturation of the endothelial types occur, it would be appropriate to follow these populations in situ during development using the markers of the 8 population identified. In particular, the perinatal detection of these cell types in their morphology / histological context will allow determining the relation between specification and maturation with kidney structure and function.

The description of the statistical test or the procedure employed to select genes for Figure 3M is missing. Regarding the same figure: the authors could further attempt a statistically grounded data binarization approach to identify cluster for which there is no evidence of regulated gene expression (i.e., only noise). This kind of binarization would make possible more appropriate claims about whether a gene is expressed or not in a cell type.

The authors state, maybe a bit casually “Additionally, molecular pathways that resolve [...] apoptosis are turned on”. If this is true and well supported it is quite a surprising finding, I would expect apoptosis to be regulated mainly at the protein cleavage/phosphorylation level, not transcriptionally. Noticeably in single-cell literature and dataset analyzed by our lab we were never able to identify an apoptosis signature on the transcriptional level. Looking at data shown in this paper the only info I found related to apoptosis is the barely significant GO term “Negative regulation of apoptosis”, I don’t think this is enough to support the claim.

The characterization of the Tbx3DeltaEC phenotype is performed very accurately. Despite the phenotype is rather mild, it is characterized in detail with evidence that encompasses histology, microstructure, and physiologic readout. Furthermore, a partial redundancy between several factors is to be expected a complex regulatory network, so the phenotype is a reasonable outcome and more important than the mild phenotype would lead to believe.

Regarding the claim that all the transcription factors have a repressive role, I regard it as a rather strong statement. I don't agree that the body evidence shown provides is enough to support the statement. For example, how are these p-values obtained? For statistics to be possible there should be biological or technical replicates but it is not specified anywhere by the authors (is N>1 also here?).

Does an only-repressive transcriptional control imply that there is a default endothelial type and in glomerular endothelium, a part of it is repressed? What is this default state? Is it one of the 6 cell

states that the author characterizes? I cannot think of another case where this only-repressive pattern was described for so many genes, it is particularly unlikely since it would also imply that all the transcription factors analyzed are the most downstream regulated and do not regulate in turn other transcription factors. If they would, for example downregulating a repressor, would lead to apparent upregulation. The authors may want to present a more detailed analysis, with an appropriate background model or further evidence for this statement.

Most of the single cell expression data is visualized through violin plots. However, almost no such a plot includes a scale on the y-axis (some not even ticks in the axes). Presented in this way data is merely qualitative. Not only authors should add the scale to the axes but, we recommend to overlay the actual data points on the violins (with some transparency and jitter for easier visualization). It is also important to indicate if the data shown is raw or normalized expression data.

It is common practice in the genomics field to provide, already at first submission, both the raw data (GEO accession) and the code that was used to produce the figures. Considering the extremely brief methodological description of the computational procedures, the lack of data and source code the paper should be considered only a partial report and cannot be thoroughly evaluated technically. We also note that the authors fail to cite some of the software they use and computational approaches inspired by previous work.

Finally, I am also wondering about data usability. Are the authors considering building a resource website or using a data visualization platform developed by the community to render this data more accessible to independent exploration? I think this is particularly important if it is recognized that the main character of this report is the on of a resource paper.

Manuscript #: NCOMMS-18-34720

Molecular determinants of nephron vascular specialization in the kidney

Point by point response to Reviewer's comments:

Response to Reviewer #1

General comments: In the manuscript entitled: Molecular determinants of glomeruli vascular specialization in the kidney, Raffi and colleagues provide comprehensive analysis of developing glomerular microvasculature through analysis of single cell RNAseq from isolated blood (non-lymphatic) endothelial cells from the kidney. In addition, the authors provide evidence for a role of *tbx3* in normal glomerular capillary formation and maturation. The paper is well-presented and provides important information and datasets to advance studies and development of therapeutics for kidney diseases – many of which affect glomeruli and their vasculature.

The following comments are for consideration by the authors:

We are grateful to the reviewer for the constructive comments. We have performed extensive additional experiments to address the remaining concerns of the reviewer.

Comment 1. Title – should be glomerular instead of glomeruli. However, the paper provides more data than just glomerular microvasculature – so would consider changing title to reflect more comprehensive data provided (which is valuable).

Response 1: As per recommendation of the reviewer, we have changed the title to:
“Molecular determinants of nephron vascular specialization in the kidney”

Comment 2. Perinatal heterogeneity – perhaps not published but not surprising given all the post-natal differentiation that occurs in adjacent structures (tubular transporters etc. aren't expressed or functional at birth but develop post-natally) – so tubular filtrate and what is reabsorbed by endothelium is very different. Could soften this statement in the introduction.

Response 2: We have softened the language for this particular aspect of the article.

Comment 3. *Tbx3* endothelial-specific knockout mice exhibit an interesting and somewhat unexpected phenotype. Typically, lower gfrs are associated with higher blood pressures – as are higher renin levels (if angiotensin 1, 2 levels are higher as a result of higher renin – one would expect a higher blood pressure). Do the authors suspect a sodium wasting phenotype in the mice to explain the bp findings (that are counterintuitive)? This would not be glomerular but downstream - typically quite distal in the nephron. A concentrating defect in the urine might also suggest this – are the mice polyuric? Is *Tbx3* also expressed in avr? Given the cre-driver is expressed in > glomerular endothelial cells, this might help explain the phenotype.

Response 3: It is possible that a sodium wasting phenotype in the *Tbx3*^{ΔEC} mice could explain the blood pressure findings. To address whether a lower GFR was associated with lower blood pressure, we added replicates to the experiment (n=4). No significant difference was found in the glomerular filtration rate following additional replicates, suggesting *Tbx3* may not affect glomerular filtration rate. More efficient deletion of *Tbx3* in the endothelial cells may also be necessary to observe glomerular filtration rate changes, as *Tbx3* is still expressed in 25-50% in the kidney vasculature. From our scRNA seq results and from antibody stainings, we do not find *Tbx3* expressed in the AVR. We mostly find *Tbx3* expression in arterioles in the kidney and lung. It is possible that *Tbx3* regulates some unknown dynamic in blood pressure homeostasis in the afferent/afferent arterioles or through the peritubular capillaries and the proximal convoluted tubule non-cell autonomously. Future studies will be necessary

to address these important questions using tools that can delete Tbx3 in particular compartments of the kidney vasculature.

Comment 4. Would remove the comment about hyperemic kidneys suggesting perfusion abnormal – soft phenotype and not diagnostic of perfusion issue – although they do look different – suggesting fixation and flushing of vasculature was not equivalent between both.

Response 4: We have removed the hyperemic kidney figures as we do not provide quantification of the phenotype.

Comment 5. It is likely there are more than 6 subpopulations of endothelial cells in the kidney – could soften this conclusion in the discussion. Findings might be limited by number of cells, stages, and might have missed some subpopulations.

Response 5: We concur with the reviewer that we might have missed rare subpopulations of kidney vasculature. We now emphasize in the revised manuscript that our goal was not to set forth a comprehensive atlas of the kidney vasculature. Our intention is to start to define the molecular signatures and functional attributes of specific key vascular zones within the kidney specifically the vasculature within the nephrons. We have now softened the language for how many populations of kidney endothelial cells we have discovered.

Comment 6. Any data to compare glomerular endos to liver sinusoidal endothelium or choriocapillaris (comparison to other published, available datasets is fine)? Given similarities between glomerular endos and some of these other fenestrated vascular beds.

Response 6: To address this, we have generated unpublished scRNAseq data of liver sinusoidal endothelium that will be submitted in another article. Sequenced sinusoidal endothelial cells manifest markers that are common of kidney peritubular capillaries, although very few genes are shared with glomerular endothelium. For example, the gene *Plvap* was found in peritubular capillaries and veins as well as liver sinusoids. This gene encodes a protein that organizes a complex called the diaphragm that exists within capillary fenestrations. Glomerular capillaries lack these structures, therefore there is a very sharp absence of this transcript in glomerular endothelium. PCA and tSNE analysis would not be of any merit in this case as glomerular endothelium and sinusoids have completely different tasks and would not cluster in adult stages. Glomerular endothelium are attached to the glomerular basement membrane and cross talk with podocytes to regulate filtration to the lumen of the nephron's Bowman's capsule, whereas liver sinusoids pass plasma into the liver interstitium, while cross talking to stellate cells. Our SCENIC results show us, in fact, that glomerular capillaries are vastly different from other endothelium found within the kidney itself, let alone other organs. Future studies may be able to discover another vascular bed that shares expression signatures and perhaps functions with glomerular capillaries, although, I believe it is beyond the scope of this study.

Comment 7. Some additional details about blood pressure measurements and histologic comparisons would be helpful – age of mice,

Response 7: The mice are 4 months of age (page 17).

--littermates

The mice used in this study were all litter mates (page 30, methods).

--gender specified

Only males were used in this study (page 30, methods).

--How were the fenestrations quantified? Etc.

Glomerular capillaries with microaneurysms were scored by the TEM core technician. Fenestrations were quantified by measuring fenestrated versus non-fenestrated lumen perimeter lengths over total perimeter length on TEM images using the program Imagej. n=5 frames, 3 mice control and 3 mice experimental.

Response to Reviewer #2:

Comment 1. Use of the term “evolution” as in the “evolution of pathways” in the introduction and several other places in the text is a questionable word choice. This is not evolution in the darwinian sense of the word. The authors also use the term “ontology”, which may be a better choice.

Response 1: We agree with the reviewer this is not the proper choice of the word and we have changed this word usage throughout the manuscript.

Comment 2. The sentence at the bottom of page 4 is problematic for several reasons. First, inclusion of the word “kidney” in the penultimate line is probably a typo. Secondly, use of the term “Unexpectedly” begs the question of what assumptions led the authors to find this result unexpected? It is not immediately obvious.

Response 2: We concur with the reviewer and we removed the word “unexpectedly” as it is not necessary to infer any assumptions. We also replaced the typo “kidney” with “liver”.

Comment 3. The sentence at the top of page 10 states that two transitory (fleeting) vascular networks are enriched in the developing cortex and a venous-like vascular bed. This implies knowledge of the location of the populations and sounds like they are distinct. What is the evidence supporting this statement?

Response 3. To address this important concern, we reanalyzed the data representing the developing vasculature. We performed pseudotime analysis using Monocle and also performed additional immunostainings. We found that the kidney is primarily composed of generic capillaries at E13 stages (Figure 2L). Pseudotime analysis and bulk RNA seq results suggest that each vascular subtype buds off these generic capillaries beginning at E14-E15 stages and into postnatal stages. We coined these early capillaries in the kidney as vascular progenitor cells. This population of endothelial cells possesses high expression of the apelin receptor (Aplnr) as validated in figure 2H. Our previous analysis produced two populations that corresponded to two stages of development rather than two disparate cell types. This is corrected and described now in the manuscript.

Comment 4. The sections dealing with the conditional mutation of Tbx3 are lacking in essential details. First, the description of the mutant mice is too sketchy. If the mice from Anne Moon, Luisa Arispe, and Ondine Cleaver have been published, references should be provided and MGI proper allele nomenclature used. A quick check of MGI indicates that there are at least 5 different Cdh5-Cre transgenes and a number of Tbx3 floxed alleles. Without detailed information on the specific alleles used, it is impossible to evaluate the quality of these mutations or to ascertain from this study that they are performing as designed. Pertinent details such as how faithfully the Cre transgene recapitulates endogenous Cdh5 expression and how efficient it is at excision of floxed alleles is essential information.

Response 4: The MGI mouse strain nomenclature and references have been added. As now clearly stated in the manuscript: Tbx3 mice (Tbx3^{tm3.1Moon})⁴², provided by Anne Moon (Weis Center for Research), were crossed with Cdh5-Cre (Tg(Cdh5-cre)7Mlia)⁴³, from Luisa Arispe (UCLA) to produce Tbx3^{fllox/fllox};Cdh5^{Cre} mice (referred to as Tbx3^{ΔEC}), and maintained as homozygous. The Cre allele was maintained in a heterozygous stage after it was bred in (Cre/+). Male littermates were used for all assays comparing control and Tbx3^{ΔEC} mice. Flk1-eGFP mice (Kdrtm2.1Jrt)⁵⁴ were kindly provided from Ondine Cleaver at UT Southwestern Medical center. R26R-Confetti mice (Gt(ROSA)26Sortm1(CAG-Brainbow2.1)Cle) were purchased from Jax and crossed to Cdh5(PAC)-CreERT2 (Tg(Cdh5-cre/ERT2)1Rha) mice from Ralph Adams. The deletion efficiency was validated using qPCR and by RNA-seq (Figures S4A and Dataset S7).

Also, it is stated that the Tbx3 mice are maintained as homozygous (sic) but it is not clear if the Cre transgene is homozygous or segregating. Do the mice examined have one or two copies of the Cre transgene?

Response 4: Mouse crossings were generated by first crossing a Tbx3^{flox/flox} stud to a Cdh5^{Cre} dam. Next, the Tbx3^{flox/+};Cdh5^{Cre/+} heterozygous male progeny were crossed to Tbx3^{flox/flox} females to generate Tbx3^{flox/flox};Cdh5^{Cre/+} progeny. These homozygous males were then only crossed to Tbx3^{flox/flox} females, therefore rendering the Cre allele heterozygous throughout all the generations of breeding. Zygosity of the Cre allele is now noted in the methods.

Did controls include mice carrying the Cre transgene? This is an important control as Cre is known to have effects on its own in some cases.

...Continued Response 4: Controls mice possessed the Tbx3^{flox/flox};Cdh5^{+/+} genotype. We omitted producing Tbx3^{flox/+};Cdh5^{Cre/+} mice because the Tbx3^{flox/+} allele (**Tbx3^{tm3.1Moon}** by Frank et al. 2012) and the Cdh5-Cre/+ alleles (Tg(Cdh5-cre)7Mlia by Alva et al. 2006) have been reported to grossly not produce any phenotypes and produce viable mice. We also performed these experiments with control mice possessing the Tbx3^{flox/flox};Cdh5^{+/+} genotype because we intended to avoid phenotypes resulting from heterozygous Tbx3 loss. Tbx3^{flox/+};Cdh5-Cre mice that might manifest 50% loss of the protein, rendering it difficult to find a statistical difference between the Tbx3^{flox/+};Cdh5-Cre full knockout mice. Therefore, we used Tbx3^{flox/flox} control mice to produce the starkest contrast between the homozygous conditional knockout mice.

Comment 5. There is no discussion of potential effects of knocking out Tbx3 in EC of other tissues, which presumably would occur. There is mention on page 21 that there is no indication of vasculopathy in other organs but no data is presented. Are the mice generally sick? How thoroughly were the mice examined for additional extra-kidney abnormalities?

Response 5: We have performed histological examination of other organs in which Tbx3 was deleted in the ECs. New histological data has been provided demonstrating no defects in heart, brain, liver, and lung tissues (n=4 control, n=4 KO after 8 months). Capillaries and arteries were not found to manifest abnormalities or aberrant depositions extracellular matrix indicating fibrosis.

Comment 6. With respect to the glomeruli in Tbx3 Δ EC mice, the proportion of abnormal glomeruli is fairly low. How does this proportion compare with normal mice and with Cre-only controls?

Response 6: The measurements we performed took into account glomeruli that look abnormal just from odd angles in cross section. To confirm these results, we performed the same analysis on control kidney cross sections. In the control kidneys we did not observe any defects in the glomeruli as observed in the Tbx3 Δ EC mice.

In the measurements of kidney function (Fig 4G-L), do the controls include mice with the Cre transgene?

Response 6: We performed these experiments with control mice with the Tbx3^{flox/flox};Cdh5^{+/+} genotype because in order to avoid phenotypes resulting from heterozygous Tbx3 loss. Tbx3^{flox/+};Cdh5-Cre mice may show 50% loss of the protein, rendering it difficult to find a statistical difference between the Tbx3^{flox/+};Cdh5-Cre full knockout mice. Therefore, we used Tbx3^{flox/flox} control mice to assess the differential phenotypes among the homozygous conditional knockout mice. Nonetheless, we find statistically different values in each experiment that correlate with the physical phenotypes produced.

Comment 7. How representative are the TEM pictures and are they taken at random? Do these represent the 75% apparently normal glomeruli or only the abnormal glomeruli? It is important to relate these TEM to the histology to understand how widespread this phenotype is and whether all the glomeruli or only the histologically abnormal ones are affected at the TEM level.

Response 7: To produce the TEM images, we utilized the Electron microscopy core at Weill Cornell Medicine. Samples were given to a core technician with specific instructions. 5 images of one kidney from three mice (15 control and 15 experimental total) were captured from cortical areas of the kidney all at the same magnification. Images for *Tbx3*^{ΔEC} mice obtained from the 17% of glomeruli that manifested microaneurysms. The lumens of the images were then quantified using Imagej. The ratio of fenestrated to non-fenestrated perimeter length per glomeruli was quantified on glomeruli clearly containing microaneurysms. Ambiguous glomeruli were omitted. A board certified pathologist aided in the identification of defective glomeruli.

Comment 8. At the bottom of page 19, what is the basis for the conclusion that the transcription factors “collaborate” to regulate a common set of targets?

Response 8: Because this phrasing is hypothetical, we have reworded the paragraph to state that the conclusion is hypothetical.

Minor points for correction:

1. There are numerous grammatical errors that need editing.

We have edited the manuscript and corrected various grammatical errors we have missed.

2. Fig. 1C the E16.5 kidney symbol must be hidden behind the blue square. Both could be visible if it were in front of the square.

We have provided a new figure and corrected this mistake.

3. Fig. 2 What age tissue is the expression validation done on? In general, the age of tissue shown in figures is not stated, e.g. what is the age/developmental stage of the human tissue in Fig. 4C (text page 16) or the mouse tissue in Fig. 4C. It should be clearly stated in the text and figure legends.

The stages are provided in the text (page 10) and figure legends.

4. Reference to Fig. 3D in the middle of page 15 seems to be incorrect as this figure is on solute transporters, not transcription factors.

This error is now corrected.

5. It should be made clear on the bottom of page 16 that *Cdh5-Cre* and *VE-cadherin-Cre* are the same thing.

This is now noted in the text.

6. Middle of page 18, Figure 4O should be 4Q.

This has been corrected.

Response to Reviewer #3:

In this work Barry et al. characterize the heterogeneity of endothelial cell populations in the kidneys. They focus on describing the vascular zonation molecularly and in characterizing the molecular determinants of the endothelial cells heterogeneity. The study, also, elucidates how these molecular differences unfold developmentally.

The main results of the paper are: (1) the definition of a specific molecular profile for 2 early and 6 mature endothelial cell populations in the kidney; (2) the description of when these differences arise developmentally; (3) the observation that gene regulation mechanisms leading to the glomerular fate is mostly repressive and (4) the characterization of the knock-out phenotype of Tbx3 a transcription factor specific to glomerular endothelial cell.

The first part of the paper has more the characteristics of a resource (Figure 1,2,3), and it is completed by a second part (Figure 4) that aims at a more in-depth experimental characterization starting from few specific findings.

I believe the paper represents an important characterization of the kidney heterogeneity to a level of detail that had escaped previous, less focused, single-cell surveys of the organ (i.e., Quake et al. 2018 “Tabula Muris”). The developmental aspect, in conjunction with the gene regulation one, is particularly interesting as it describes the timing of kidney endothelial specification to unprecedented detail. We would like to recommend this work for publication in Nature Communications as long as the authors make substantial changes to the manuscript, significantly extend and improve the RNA-seq analysis and its statistical rigor, placing more care in the conclusions reached from the data. We also suggest only a few experimental extensions to complete and clarify the data the authors present.

The part of the paper that I am more concerned about is the bulk transcriptomics analysis of the endothelium of different organs as compared with the kidney endothelium (Paragraph “Molecular profiling of kidney endothelium” and Figure 1A-B, S1A-D). There are several problems with this part of the paper:

Response to the general Comments: We are grateful to the reviewer for constructive comments. Based on the insightful recommendation of the reviewer, we have now performed additional in depth computational analyses to improve the representation and inferences from both bulk and single cell RNA sequencing data.

Comment 1: First and foremost, it seems (the author do not state otherwise) that all the data has no biological or technical replicates, in other words, N=1 !!! This fact alone takes a heavy toll on the statistical validity of the results, and if really N is equal 1, then no statistical inference can be made from this part of the data.

Response 1: We have clarified this very important concern. We have now summarized the combinatorial approach we employed to generate replicates in the figure legends. Isolation of pure populations of endothelial cells from early embryonic stages are difficult to obtain. Around forty embryos were pooled from six pregnant mice for E13-E15 stages each, for example. Therefore, we performed the experiments related to the day 13 to 17 of embryonic development once to annotate genes, as standard practice. However, because of high similarity of the molecular signatures among various organs and intra-organ vascular similarities during these early stages of the development, we were able to combine these samples to generate as a whole a fetal vascular reference point. Indeed, we do agree that statistical inference and differential expression cannot be obtained for individual genes with n = 1 samples used for gene annotation. However, we do not draw any statistical inference from this dataset using individual genes. In our clustering analysis, as a standard practice, we first summarized gene expression values by taking the mean value between replicates when available. *This indicated (Figure 1B) that bulk gene expression patterns of the early embryos were highly similar.* Thus,

we grouped all embryonic stages together when looking for differential gene expression between embryonic and adult stages. The adult kidney was sequenced at n=4 biological replicates. RNAseqs involving an experimental manipulation (i.e. gene knockdown or gene over-expression) in which we expect some variation due to technical manipulation absolutely require at least n = 3. In instances where we were examining differential gene expression between two conditions (i.e. between Tbx3 control and knockout mice, and between control and transcription factor overexpression), all samples were sequenced at least 3 times (n=3).

Comment 2: The comparison between a general “kidney endothelial cell” and the ones from other organs is conceptually flawed, especially if considering the data that the authors have at hand. The authors show, later in the paper, that there are important differences between endothelial cell types in the kidney. So that comparing a pool of different endothelial cells RNA mixed together results in an artefactual cell expression pattern that does not exist in any real cell but is the average of them all. The entire analysis is influenced by this averaging, so statements like “kidney endothelial cells possess distinct expression pattern” are imprecise, the inferences here is on a heterogeneous pool difficult to analyze and interpret and therefore at the mercy of artefactual results (e.g. a case of Simpson’s paradox). We also note that in this analytic scenario even a slight change in the composition of the subtypes might influence the results of the analysis.

Response 2: We have considered the constructive reviewer comment and we show that inclusion of bulk RNA-seq could be informative to define the results obtained with single cell RNA-seq (scRNA-seq). Even though the results are pooled between capillaries, veins, and arteries, the goal is to identify the specific genes that exist in one organ and not the other organs. Because the pooling effect can cause artificial ratios of RNA that do not actually reflect ratios of capillaries vs arteries, etc., this data set is primarily useful to observe transcripts which are unique to particular organs, but not differences in particular transcript levels. This data set is useful depending on the questions being interrogated. For example, searching for putative genes expressed in the kidney vasculature, such as Urea transporters, then one can find this transcript only in the kidney. However, in terms of more broadly expressed genes such as Pecam1 (CD31), a vascular marker that could be expressed at higher level in the kidney, a comparison of this commonly expressed genes among various organs would not be informative because of the error in transcript ratios between vascular subtypes (arteries, veins and capillaries). Accordingly, in this data set, we only analyzed genes which were unique to the kidney or prominently differentially expressed.

Comment 3: There is a significant gap between the data interpretation provided by the authors and what the data presented is suggesting. The PCA analysis shown in figure 1 is overinterpreted and the authors stretch too many conclusions out of it. Despite the fact that the authors try to lead the eye of the reader by drawing circles around points the differences that they comment on are not so clear from the data. Furthermore, we noticed that the sample coordinates on PC1 are not zero centered and have an unusual range, are the authors using some other sample in the analysis and not showing it on the plot? Also being the percentage of variance explained by PC1 very high (small in PC2) and the values of PC1 all positive lead me to conclude that the difference between the data point is just linear scaling of the expression profile. This would be in sharp contrast to what the authors claim and suggest a mere difference in detection levels or cell size.

Response 3: Principal component analysis was used merely as a clustering technique. The first principal component explained a majority of variation in the dataset because the data was derived from cells of all the same cell types with highly similar gene expression pattern. To address reviewer’s concern, we have replaced the PCA clustering analysis with affinity propagation clustering (APC). APC is a deterministic clustering method that will determine numbers of clusters and cluster identities in a solely data-driven process. No arbitrary boundaries for cluster make-up need to be made using this technique.

Comment 4: The authors ignore the availability of at least a dozens of single-cell datasets in the literature that incidentally includes endothelial cells of one or more organs, and that could be used to make a better analysis than the one allowed from this bulk experiments. In summary, I believe the evidence in this part of the paper is confusing, and the analysis does not add any relevant information to the main story. I would therefore strongly recommend the authors to completely remove this analysis from the paper and start directly from the single cell RNA-seq data. The alternative would be the substitution of their analysis with an improved or different dataset that can provide more compelling evidence of their claims. The single-cell data collected is adequate to most of the claims made by the authors. Maybe it could have even be used to make further or more precise claims.

Response 4: Although the publicly available single-cell data sets are valuable in terms of divulging what transcript is expressed in what type of blood vessel, they do not give a precise purview of all the genes that are uniquely expressed in the various vascular beds. Even though the bulk RNA seq analysis does not accurately reflect the ratios of genes expressed in each type of vessel, it has complete coverage of all the transcripts that exist. The majority of scRNA seq data sets that include a diminishingly small number of endothelial cells are derived from droplet based technologies such as 10x Chromium. Although these technologies are instructive at finding new cells types and transcripts in high abundance, they are very poor at capturing low level transcripts and even canonical markers of various cell types. Only 50% of transcripts that are expressed are captured in many cases. Therefore, bulk analysis is important as a control to access all the potential transcripts that exist in a pool of cells. As a follow up, scRNA-seq analysis can be used to pull apart different cell types in the pool. Even if we did extract differential analysis of genes expressed between scRNA-seq samples from other studies, this also suffers from the same shortcomings. The cell numbers are going to be different from the differences and errors in the digestion technique, and since only 50-80% of the transcripts are possibly captured, there's a possibility that several genes could appear differentially expressed when they were in fact not captured. Therefore, for these particular studies, it is helpful to use both bulk and scRNA-seq approaches in tandem.

Comment 5: Notably, the authors do not propose a model for the change in the heterogeneity of the endothelial pool and specification observed during development. This cell state transition is an important finding in the economy of the paper and the authors should try to complete its analysis and description. Can the data be used to build a model of the lineage determination starting from the 2 early types and ending in 6 mature types? For example, the data should allow determining the structure of the lineage either by branching pseudotime analysis (monocle 3), optimal transport analysis (Waddington OT) or RNA velocity analysis (velocyto) to try to come up with a model of this developmental progression. Coming up with a putative lineage branching structure would also allow a more detailed analysis of the specification of endothelial cells. We strongly encourage the authors to at least propose a putative qualitative model of this developmental process based on the evidence available. What is the most likely progeny of each adult clusters? Is it possible to determine which transitions are allowed? Which cells are proliferating and which one postmitotic? If the authors did not develop these analyses because they are convinced that the data is insufficiently detailed or high quality to fit this model, they should consider sampling more cells.

Response 5: As suggested by the reviewer, we have performed pseudotime analysis via Monocle to ascertain the hypothetical developmental trajectory of vascular subtypes in the kidney that develop from vascular progenitor cells. As shown in the revised manuscript, the pseudotime trajectory predicts capillaries in the developing kidney can remain venous and prescribe into larger venous structures (Nr2f2⁺, Cryab⁺, Plvap⁺) (Figure 2E, S2B), maintain as capillary progenitors (Aplnr⁺, Igfbp5⁺, Plvap⁺) (Figure 2B, 2D-E, S2C), or completely shift into arterial blood vessels (Fbln5⁺, Jag1⁺ Sox17⁺, Cxcl12⁺, Gja5⁺) (Figure 2B, 2F, S2D-E). Progenitor capillaries that don't prescribe to either of these fates can

branch off as glomerular capillaries (Lpl^+ , $Ehd3^+$, $Sema5a^+$, etc.) (Figure 2C, 2G) in association with efferent arterioles or the DVR ($Slc14a1^+$, $Aqp1^+$) (Figure 2C, S2F-G), which is successive to the juxtaglomerular efferent arteriole. Remaining capillaries lose $Aplnr$ and $Nr2f2$ expression and maintaining expression of several genes enriched in mature capillaries such $Igfbp5$ and $Plvap$ (Figure 2C, 2E, S2B-C). These results are consistent with studies which show arterial blood vessels are among the first blood vessels to become specified from generic capillaries in the early kidney (Daniel et al. *Angiogenesis* 2018 Aug;21(3):617-634). Results in this study confirm that other subvascular structures primarily develop subsequently to E15 stages.

Comment 6: Finally, the two early subtypes (cluster 1 and 2) define transcriptionally should be anchored to cell populations in the tissue by means of RNA-scope or smFISH (better than IH because some of the genes may not be translated at a higher level yet). In this way, the gene expression patterns can be connected to the histology and cell morphology offering a more precise picture of what these populations are.

Response 6: After further careful analysis we found that the 2 early cluster subtypes can be clustered together and were more the gradient of expression across consecutive stages. This cluster is well defined by expression of the marker $Aplnr$, which is also a marker of vascular progenitor cells in the heart as well (Sharma et al. *Dev Cell* 2017).

Comment 7: Furthermore, concerning the finding that a perinatal specification and postnatal maturation of the endothelial types occur, it would be appropriate to follow these populations in situ during development using the markers of the 8 population identified. In particular, the perinatal detection of these cell types in their morphology / histological context will allow determining the relation between specification and maturation with kidney structure and function.

Response 7: Although the markers of each vascular subtype appear to be upregulated on a RNA level perinatally, overall, the genes that define each vascular subtype become expressed much earlier in development. As shown by Daniel et al. 2018, the kidney only begins to develop vascular heterogeneity around E15, when glomeruli, arterioles, and the vasa recta become specialized. This was corroborated by our RNA-seq results of markers at these stages (Figure 3A). Therefore, we focused on validating expression of adult markers at E15.

Comment 8: The description of the statistical test or the procedure employed to select genes for Figure 3M is missing.

Response 8: Most of the transporters expressed were not captured in the single-cell analysis (based on the bulk RNA seq analysis). The few ones that were are mostly shown here and were arbitrarily chosen, based on information published defining functional contribution of these genes to various physiological processes. For the readers to investigate gene inquires in the single-cell RNA seq data set, we have made the data publically available through gene expression omnibus (GEO), with an embargo on data release until publication. Accession numbers are provided in the methods.

Comment 9: Regarding the same figure: the authors could further attempt a statistically grounded data binarization approach to identify cluster for which there is no evidence of regulated gene expression (i.e., only noise). This kind of binarization would make possible more appropriate claims about whether a gene is expressed or not in a cell type.

Response 9: A statistical approach for gene expression thresholds is provided in the supplementary Data set 5. This spread sheet provides a p value for expression significance for the most enriched genes in each cluster (including captured transporters) and also provides the expression level in comparison to the other clusters.

Comment 10: The authors state, maybe a bit casually “Additionally, molecular pathways that resolve [...] apoptosis are turned on”. If this is true and well supported it is quite surprising finding, I would expect apoptosis to be regulated mainly at the protein cleavage/phosphorylation level, not transcriptionally. Noticeably in single-cell literature and dataset analyzed by our lab we were never able to identify apoptosis signature on the transcriptional level. Looking at data shown in this paper the only info I found related the apoptosis is the barely significant GO term “Negative regulation of apoptosis”, I don’t think this is enough to support the claim.

Response 10: The pathway enrichment analysis withdrew “negative regulation of apoptosis” and “negative regulation of developmental processes” through recognizing the genes MPO and SNCA. These genes were significantly differentially expressed in endothelial cells between E17 other stages of development and are associated with these two pathway titles (provided by the GO tool used). The GO tool developed these pathway titles supposedly through a pool of published work. Via recognition of these two genes, the pathway enrichment analysis hypothesizes that negative regulation of apoptosis developmental processes may be occurring. Therefore, when we stated “Prior to birth (E17) [...] molecular pathways that resolve developmental programs and apoptosis are turned on”, it is based on a hypothesis of the enrichment of genes associated with these GO terms. To obtain of an all-inclusive pathway analysis we grouped additional embryonic stages together and no longer possess this term in any figures.

Comment 11: The characterization of the Tbx3DeltaEC phenotype is performed very accurately. Despite the phenotype is rather mild, it is characterized in detail with evidence that encompasses histology, microstructure, and physiologic readout. Furthermore, a partial redundancy between several factors is to be expected a complex regulatory network, so the phenotype is a reasonable outcome and more important than the mild phenotype would lead to believe. Regarding the claim that all the transcription factors have a repressive role, I regard it as a rather strong statement. I don't agree that the body evidence shown provides is enough to support the statement. For example, how are these p-values obtained? For statistics to be possible there should be biological or technical replicates but it is not specified anywhere by the authors (is $N > 1$ also here?). Does an only-repressive transcriptional control imply that there is a default endothelial type and in glomerular endothelium, a part of it is repressed? What is this default state? Is it one of the 6 cell states that the author characterizes? I cannot think of another case where this only-repressive pattern was described for so many genes, it is particularly unlikely since it would also imply that all the transcription factors analyzed are the most downstream regulated and do not regulate in turn other transcription factors. If they would, for example downregulating a repressor, would lead to apparent upregulation. The authors may want to present a more detailed analysis, with an appropriate background model or further evidence for this statement.

Response 11: The claim that all the transcription factors have a repressive role is a hypothesis we developed because a large number of genes differentially expressed in glomerular endothelial cells become downregulated after over expression of Tbx3, Gata5, Prdm1, and Pbx1 in HUVECs (human umbilical vein endothelial cells) in vitro. Supporting this hypothesis, there is a body of literature suggesting these factors may act as repressors of transcription. Because we do not provide more follow up experiments to cement this claim, we have softened the language to infer that this statement is a hypothesis.

Comment 12: Most of the single cell expression data is visualized through violin plots. However, almost no such a plot includes a scale on the y-axis (some not even ticks in the axes). Presented in this way data is merely qualitative. Not only authors should add the scale to the axes but, we recommend to overlay the actual data points on the violins (with some transparency and jitter for easier visualization). It is also important to indicate if the data shown is raw or normalized expression data.

Response 12: We have added scales and ticks for the y-axis of the violin plots. We omitted the data points primarily because they blocked the violin, which represents the location for the bulk of data points. This made the graphs somewhat difficult to view, especially in the smaller plots. The data represents the normalized data and we have updated figure legends accordingly.

Comment 13: It is common practice in the genomics field to provide, already at first submission, both the raw data (GEO accession) and the code that was used to produce the figures. Considering the extremely brief methodological description of the computational procedures, the lack of data and source code the paper should be considered only a partial report and cannot be thoroughly evaluated technically. We also note that the authors fail to cite some of the software they use and computational approaches inspired by previous work.

Response 13: Citations for software were provided in the methods section. We apologize for the oversight and we have properly cited software in the results section. We have added a section in the methods section for source code availability, and data has been deposited onto Gene Expression Omnibus (GEO). Dataset accession numbers have been added to the methods sections.

Comment 14: Finally, I am also wondering about data usability. Are the authors considering building a resource website or using a data visualization platform developed by the community to render this data more accessible to independent exploration? I think this is particularly important if it is recognized that the main character of this report is the one of a resource paper.

Response 14: We are grateful to the reviewer for this important recommendation. Building such a website is important and upon publication of this manuscript we can hopefully secure sufficient funding to jumpstart this website. Meanwhile, we have uploaded the normalized expression matrices to the GEO website.

Reviewers' comments:

Reviewer #1 (Remarks to the Author):

This is an important description and provision of datasets for endothelial cells of the developing and mature kidney - underscoring the heterogeneity of these cells.

The authors have responded to my prior questions. One remaining point: profound glomeruli defects referred to in Fig 4D appears overstated. It is not clear from the description whether there are more glomerular morphogenetic defects at earlier timepoints - however, at 4 months, the defects are quite subtle.

Reviewer #2 (Remarks to the Author):

The authors have answered all my points and clarified the alleles used in the Tbx3 deletion study. However, I am not completely satisfied with all the explanations.

With respect to the lack of a Cre-containing control, they have explained their reasoning but the control is still an important one. If they are unable to provide such a control there should be a statement in the discussion with a caveat that they have not controlled for the presence of Cre alone and a justification as to why they think they can ignore it.

Similarly, the authors now provide gross histology showing no overt abnormalities in vasculature in other organs in the Tbx3-EC knockout, but provide no discussion as to why there should be a phenotype only in one organ with an EC knockout. Surely this deserves a bit more discussion as Tbx3 expression is quite widespread.

Reviewer #3 (Remarks to the Author):

Comment to Response 1: Frankly, I did not understand what the authors mean with a requirement for gene annotation. Gene annotation of the mouse genome and transcript variance have been well

annotated by previous efforts. Furthermore, detailed splicing variant calling is not necessary for good single cell RNA-sequencing counting, so the all statement does not make much sense to me.

On the other point, I agree with the authors that the data can be used to prove that "bulk gene expression patterns of the early embryos were highly similar"

Comment to Response2: The point on this response can be made slightly more explicit in the text. But I think now the claims based on bulk data are more moderate.

Comment to Response3: For the author's interest I would like to point out to them that PCA (aka SVD) is not a clustering technique but it is actually a change of coordinates that allow better visualization of the data.

Anyways, showing this point with affinity propagation is supportive of the author's interpretation. Still, the PCA lead me to a different conclusion, there must have been some coding bug. Maybe having the Pca plots (after correcting the problems that I pointed out before) in supplementary might help the interested reader to understand better the variation in the data.

Comment to Response 4: I remain in disagreement on this point with the authors. I can understand if the authors wish to maintain the main structure of their paper, and that over a long project historical decision of applying an older technique vs a newer one are real-world choices to be taken in consideration. Furthermore, I appreciate the changes in the main text towards putting less emphasis on this part of the analysis. So I consider the point not major anymore but I would warn the authors against using a bunch very weak claims to defend their choices. The choice Bulk vs single-cell RNA seq for the present application remains a no brainer, I don't think any of the statement they make about depth and detection levels is true.

Comment to Response 5: It is good to see that now the authors use the data to present a model for the developmental transition observed. Note: Since this is not my field of biological expertise I cannot judge on the credibility of the model in the light of previous literature.

Comment to Response 6: We are happy that our comment led the authors to a more critical analysis and to conclude that the two population constitute just a gradient.

Comment to Response 7: Credible for me, but I am not an expert, so maybe the other reviewers might want to validate.

Comment to Response 8: it is ok as soon as clearly stated in the text

Comment to Response 9: This was originally just an invitation to improve the paper on a statistical side with more sophisticated modeling, the authors stayed with what they started. The authors did not take advantage of the opportunity... but ok I guess.

Comment to Response 10: What troubled me seems gone. Thank you for the authors for the explanation, the evidence was poorly supported as far as I understand as lead by a couple of genes.

Comment to Response 11: Ok

Comment to Response 12: is there a reason why values start from 1? I think there is still a mistake and values should be 0.

Comment to Response 13: This is still not dealt with. The code should be available for download on Github as it is essential for reproducibility. GEO entry should be made accessible to reviewers during the review.

Comment to Response 14: Both hosting and building these resources is nowadays free, using many packages created by the community, these visualizations can be created by non-experts. We encourage the authors to make this tool available before publication date.

Point by Point Response to the Reviewers' comments:

Manuscript NCOMMS-18-34720B:

Molecular determinants of nephron vascular specialization in the kidney

Response to Reviewer #1:

This is an important description and provision of datasets for endothelial cells of the developing and mature kidney - underscoring the heterogeneity of these cells. The authors have responded to my prior questions. One remaining point: profound glomeruli defects referred to in Fig 4D appears overstated. It is not clear from the description whether there are more glomerular morphogenetic defects at earlier timepoints - however, at 4 months, the defects are quite subtle.

Response: We have revised the manuscript to give a more accurate description of the glomerular phenotypes observed in the Tbx3 conditional knockout mice.

Response to Reviewer #2:

The authors have answered all my points and clarified the alleles used in the Tbx3 deletion study. However, I am not completely satisfied with all the explanations.

With respect to the lack of a Cre-containing control, they have explained their reasoning but the control is still an important one. If they are unable to provide such a control there should be a statement in the discussion with a caveat that they have not controlled for the presence of Cre alone and a justification as to why they think they can ignore it.

Response: To provide confidence that the Cre line does not produce any off-target phenotypes observed in the Tbx3 conditional knockout in the glomeruli endothelium, we generated control mice in which heterozygote VE-cadherin-Cre ($CDH5^{Cre/+}$) was crossed to homozygous floxed Tbx3 gene mice. This cross resulted in the generation of $Tbx3^{flox/+};Cdh5^{+/+}$ and $Tbx3^{flox/+};Cdh5^{Cre/+}$ mice, in which we do not expect to observe any glomeruli defects. Thus, if we unexpectedly observe glomeruli defects in these control mice, such as aberrant fibrosis, microaneurysms, or hypoplasia detected previously in homozygous endothelial Tbx3-deficient mice or any other kidney defects, could be due the off-target effects of the Cre line. As we described for the timeline of studying homozygous $Tbx3^{flox/flox};Cdh5^{Cre/+}$ mice, these control mice were aged four months, and subsequently their kidneys were rigorously analyzed by histological analyses for any defects with the aid of a board certified pathologist.

In these control $Tbx3^{flox/+};Cdh5^{+/+}$ and $Tbx3^{flox/+};Cdh5^{Cre/+}$ mice, we were not able to find any defective glomeruli with aberrant fibrosis, microaneurysms, or hypoplasia (**Figure 1 below**; $Cre^{+/+}$ n=3 mice, n=36 sections, n=7498 glomeruli; $Cre^{+/+}$ n=3 mice, n=36 sections, n=7546 glomeruli). Based on these results we conclude that the observed defects in the glomeruli of the homozygous $Tbx3^{flox/flox};Cdh5^{Cre/+}$ mice is not due to Cre toxicity and is the direct cause of homozygous Tbx3 deficiency, specifically in the endothelial cells.

Similarly, the authors now provide gross histology showing no overt abnormalities in vasculature in other organs in the Tbx3-EC knockout, but provide no discussion as to why there should be a phenotype only in one organ with an EC knockout. Surely this deserves a bit more discussion as Tbx3 expression is quite widespread.

Response: By single-cell RNA seq, we show that Tbx3 expression is mostly restricted and is primarily observed in the glomerular capillaries. Because of this restricted expression, it does not manifest wide-spread role in cell physiology and may not heavily impact cell phenotypes throughout the other organs where it is not expressed. In our dataset and other public datasets (data not shown), Tbx3 is not expressed in other organ-specific blood vessels. We find it only expressed in arterioles in the lung and glomerular capillaries in the kidney. If at steady state conditions, Tbx3 gene loss does not greatly impact the function of lung arterioles, thus it may not be surprising that its loss in the vasculature has phenotypes under steady state conditions are primarily detected in the kidney. These remarks have now been added to the discussion.

Figure 1. The Cdh5-Cre allele does not generate kidney phenotypes in Tbx3-heterozygote or wild type mice.

Tbx3^{flox/+};Cdh5^{+/+}

Tbx3^{flox/+};Cdh5^{Cre/+}

Response to Reviewer #3:

Comment to Response 1: *Frankly, I did not understand what the authors mean with a requirement for gene annotation. Gene annotation of the mouse genome and transcript variance have been well annotated by previous efforts. Furthermore, detailed splicing variant calling is not necessary for good single cell RNA-sequencing counting, so the all statement does not make much sense to me. On the other point, I agree with the authors that the data can be used to prove that "bulk gene expression patterns of the early embryos were highly similar."*

Response: We are grateful to the reviewer for the proper interpretation of our results.

Comment to Response 2: *The point on this response can be made slightly more explicit in the text. But I think now the claims based on bulk data are more moderate.*

Response: We agree with this assessment

Comment to Response 3: *For the author's interest I would like to point out to them that PCA (aka SVD) is not a clustering technique but it is actually a change of coordinates that allow better visualization of the data.*

Anyways, showing this point with affinity propagation is supportive of the author's interpretation. Still, the PCA lead me to a different conclusion, there must have been some coding bug. Maybe having the Pca plots (after correcting the problems that I pointed out before) in supplementary might help the interested reader to understand better the variation in the data.

Response: We apologize for the confusion in our reply. We do understand that PCA is a dimensionality reduction technique. However, in this instance, we are using PCA merely as a method to group samples post-dimensionality reduction based on similarity in gene expression. This is a very common application of PCA. We have replaced this data with a true clustering technique, affinity propagation clustering, in response to the reviewer's comment. The PCA in the original submission was performed according to the standard practice. The coordinates are zero centered. Because these samples all are derived from the same cell type (endothelial cells), the gene expression is highly similar for the majority of the genes. This explains why the first principal component captures the majority of the variation in the data. In our paper, we focus on the subset of genes that are differentially expressed using a combination of single cell and bulk RNA sequencing to decipher sources of vascular specification between subsets of endothelial cells

Comment to Response 4: *I remain in disagreement on this point with the authors. I can understand if the authors wish to maintain the main structure of their paper, and that over a long project historical decision of applying an older technique vs a newer one are real-world choices to be taken in consideration. Furthermore, I appreciate the changes in the main text towards putting less emphasis on this part of the analysis. So I consider the point not major anymore but I would warn the authors against using a bunch very weak claims to defend their choices. The choice Bulk vs single-cell RNA seq for the present application remains a no brainer, I don't think any of the statement they make about depth and detection levels is true.*

Response: As we have validated the single-cell RNA data then we are assured that the data presented has strong scientific rigor without bulk RNA sequencing

Comment to Response 5: *It is good to see that now the authors use the data to present a model for the developmental transition observed. Note: Since this is not my field of biological expertise I cannot judge on the credibility of the model in the light of previous literature.*

Response: we assure the reviewer these developmental trajectories report on the proper transition of various zone specific vascular cells.

Comment to Response 6: *We are happy that our comment led the authors to a more critical analysis and to conclude that the two population constitute just a gradient.*

Response: We concur with this assessment

Comment to Response 7: *Credible for me, but I am not an expert, so maybe the other reviewers might want to validate.*

Response: Validation of these points are beyond the scope of this manuscript.

Comment to Response 8: *it is ok as soon as clearly stated in the text*

Response: We ascertain that these points were clearly described and the means by which each gene was chosen is now stated in the figure legend.

Comment to Response 9: *This was originally just an invitation to improve the paper on a statistical side with more sophisticated modeling, the authors stayed with what they started. The authors did not take advantage of the opportunity... but ok I guess.*

Response: Validation of these points are beyond the scope of this manuscript.

Comment to Response 10: *What troubled me seems gone. Thank you for the authors for the explanation, the evidence was poorly supported as far as I understand as lead by a couple of genes.* **Response:** Thank you for your understanding.

Comment to Response 11: Ok

Comment to Response 12: *is there a reason why values start from 1? I think there is still a mistake and values should be 0.*

Response: The choice of starting the cluster numbering on 0 or 1 is based on preference and aesthetics.

Comment to Response 13: *This is still not dealt with. The code should be available for download on Github as it is essential for reproducibility. GEO entry should be made accessible to reviewers during the review.*

Response: At the request of the reviewer, we have added an additional data table containing the raw counts used as input into the single-cell RNA sequencing pipelines. All normalized bulk RNAseq expression values are made available to the authors as supplemental data tables. The GEO submission contains the raw data and is embargoed until publication. We have made available the code for processing single cell RNA sequencing as supplementary data files.

Comment to Response 14: *Both hosting and building these resources is nowadays free, using many packages created by the community, these visualizations can be created by non-experts. We encourage the authors to make this tool available before publication date.*

Response: We have reached out to experts to assist in building these websites. We expect to be able to host an easily accessible reference site in the very near future.

REVIEWERS' COMMENTS:

Reviewer #2 (Remarks to the Author):

My concerns have been satisfactorily addressed.

Reviewer #3 (Remarks to the Author):

No further comments.